# A multi-platform reference for somatic structural variation detection

## Graphical abstract

## Authors

Jose Espejo Valle-Inclan,
Nicolle J.M. Besselink, Ewart de Bruijn, ...,
Remond J.A. Fijneman,
Wigard P. Kloosterman, Edwin Cuppen

## Correspondence

wigardkloosterman@gmail.com (W.P.K.),
e.cuppen@hartwigmedicalfoundation.nl
(E.C.)

## In brief

Structural variation (SV) is part of the cancer genome mutational landscape driving tumor characteristics. Due to its nature, detection of somatic SVs is complicated and unresolved. DNA sequencing technology and data analysis tools are developing rapidly. Valle-Inclan et al. describe the generation of a carefully curated and validated somatic SV truth set based on the COLO829 cell line that can be used as a reference for benchmarking existing and novel somatic SV detection techniques.

## Highlights

- Whole-genome analyses using multiple state-of-the-art techniques

- Renewable COLO829 tumor-normal cell line pairs

- Generation of a carefully curated and validated somatic SV truth set

- Publicly available benchmark dataset for sequencing technology and analysis software

Espejo Valle-Inclan et al., 2022, Cell Genomics 2, 100139
June 8, 2022 © 2022 The Author(s).

CellPress

## Resource

# A multi-platform reference for somatic structural variation detection

Jose Espejo Valle-Inclan,[1] Nicolle J.M. Besselink,[1] Ewart de Bruijn,[2] Daniel L. Cameron,[2,3] Jana Ebler,[4] Joachim Kutzera,[1] Stef van Lieshout,[2] Tobias Marschall,[4] Marcel Nelen,[5] Peter Priestley,[2] Ivo Renkens,[1] Margaretha G.M. Roemer,[6] Markus J. van Roosmalen,[1] Aaron M. Wenger,[7] Bauke Ylstra,[6] Remond J.A. Fijneman,[8] Wigard P. Kloosterman,[1,*] and Edwin Cuppen[1,2,9,*]

[1]Center for Molecular Medicine and Oncode Institute, UMC Utrecht, Utrecht, the Netherlands
[2]Hartwig Medical Foundation, Amsterdam, the Netherlands
[3]Bioinformatics Division, Walter and Eliza Hall Institute of Medical Research, Melbourne, VIC, Australia
[4]Institute for Medical Biometry and Bioinformatics, Medical Faculty, Heinrich Heine University Düsseldorf, Düsseldorf, Germany
[5]Department of Human Genetics, Radboud UMC, Nijmegen, the Netherlands
[6]Department of Pathology, Amsterdam UMC, Vrije Universiteit Amsterdam, Cancer Center Amsterdam, Amsterdam, the Netherlands
[7]Pacific Biosciences, Menlo Park, CA, USA
[8]Department of Pathology, Netherlands Cancer Institute, Amsterdam, the Netherlands
[9]Lead contact
*Correspondence: wigardkloosterman@gmail.com (W.P.K.), e.cuppen@hartwigmedicalfoundation.nl (E.C.)

## SUMMARY

Accurate detection of somatic structural variation (SV) in cancer genomes remains a challenging problem. This is in part due to the lack of high-quality, gold-standard datasets that enable the benchmarking of experimental approaches and bioinformatic analysis pipelines. Here, we performed somatic SV analysis of the paired melanoma and normal lymphoblastoid COLO829 cell lines using four different sequencing technologies. Based on the evidence from multiple technologies combined with extensive experimental validation, we compiled a comprehensive set of carefully curated and validated somatic SVs, comprising all SV types. We demonstrate the utility of this resource by determining the SV detection performance as a function of tumor purity and sequence depth, highlighting the importance of assessing these parameters in cancer genomics projects. The truth somatic SV dataset as well as the underlying raw multi-platform sequencing data are freely available and are an important resource for community somatic benchmarking efforts.

## INTRODUCTION

Structural genomic variations (SVs) form a major class of somatic genetic variation in cancer genomes,[1,2] involving dozens to thousands of somatic SVs with varying size distribution and patterns.[2] While some SVs represent simple deletions, others tend to be complex, involving multiple breakpoints across a relatively short genomic interval. For example, chromothripsis is a form of complex SVs frequently observed in cancer genomes,[3,4] resulting from aberrant chromosome segregation or telomere dysfunction.[5,6] Other types of complex SVs involve oncogene amplifications arising from breakage-fusion-bridge cycles.[2,7,8] SVs in cancer genomes may promote cancer development through a variety of mechanisms, such as oncogene activation through gene fusions, disruption of tumor-suppressor genes, or by affecting gene regulation.[9,10] Oncogenic fusion genes resulting from somatic SVs form important targets for cancer drugs, and somatic SVs may form neo-antigenic targets for immunotherapies,[11] demonstrating the relevance of accurate somatic SV detection for personalized cancer treatment.[10,12]

While classical karyotyping and fluorescence *in situ* hybridization (FISH) analyses have been instrumental in systematic copy number analyses in tumor samples,[10,12] these technologies provide limited resolution or do not allow for comprehensive genome-wide analysis and are thus unable to resolve the complete spectrum of SV events. Most of our knowledge on genome-wide, high-resolution SVs in cancer genomes stems from the analysis of short-read, whole-genome sequencing, which is currently the only scalable and cost-efficient technology for high-resolution, genome-wide cancer genome analysis.[2,13] Although short reads are effective for detection of simple SV breakpoints in non-repetitive regions of the genome, the interrogation of complexly rearranged regions or the detection of SV breakpoints in low-complexity genomic regions may require other sequencing techniques or targeted approaches.[14] For example, long-insert, mate-pair sequencing has proven a valuable strategy for genome-wide mapping of somatic SVs,[15,16] and single-cell template strand sequencing enables the detection of copy number variants and copy neutral structural variants.[17] Furthermore, long-read sequencing methods and linked-read approaches, such as Pacific Biosciences, Oxford Nanopore, and 10× genomics, provide a promising alternative for the detection of SVs. Initial studies have shown that long-read, single-molecule sequencing can greatly improve detection

of germline SVs.[18–21] Similarly, recent work has demonstrated the advantage of long-range sequence information for identification of SVs in cancer genomes, such as cancer-gene amplifications and gene-fusion events.[8,22–24]

A major limitation of studies on cancer SVs is the lack of comprehensive ground-truth, genome-wide somatic SV datasets, including all types and sizes of somatic structural aberrations. Such truth sets can form a resource for benchmarking sequencing and analysis approaches as well as for evaluating detection problems related to intratumor heterogeneity and tumor purity. Truth sets have been established for germline SVs[21,25] or somatic single-nucleotide variants (SNVs).[26] However, attempts at benchmarking somatic SVs have only been performed by using *in silico* simulated data[27,28] or mouse data.[29]

We addressed this caveat by generating a multi-platform, short-read, long-read, linked-read sequencing and optical mapping dataset for the COLO829 melanoma cell line and the paired COLO829BL lymphoblastoid reference cell line. These cell lines were derived from a male individual and have been used before to establish somatic SNV and copy number alteration (CNA) truth sets.[26,30,31] By cross-platform comparison and extensive validation and curation, we define a truth set of 68 somatic SVs in COLO829. We evaluated the completeness of this validated SV truth set and demonstrated its use to study the effect of tumor purity and sequencing coverage variation on the accuracy of somatic SV calling. We believe this somatic SV truth set to be of broad value for benchmarking and quality control of large-scale, cancer-genome sequencing studies, which are currently undertaken in research and the clinic.

## RESULTS

### Multi-platform, genome-wide analysis of the COLO829 tumor-normal melanoma cell line pair

In this study, we aimed to obtain a high-quality validated set of somatic structural variants. We cultured COLO829 and the corresponding lymphoblastoid cell line (COLO829BL) according to standard conditions (STAR Methods). A large batch of cells expanded from one original vial directly obtained from the ATCC cell line repository was used for DNA isolation and subsequent genomic analysis using five different technology platforms: Illumina HiSeq X Ten (ILL), Oxford Nanopore Technologies (ONT), Pacific Biosciences (PB), 10× genomics (sequenced on Illumina NovaSeq) (10X), and Bionano Genomics Saphyr optical mapping (BNG) (STAR Methods).

The sequencing and optical mapping data were analyzed with respect to the reference human genome (GRCh37) using alignment methods specific for each technology (STAR Methods). From the combined short- and long-read sequencing data of the COLO829 sample, we obtained a total average base coverage of 235×. The BNG data generated an additional physical coverage of 218×. For the COLO829BL control cell line, a combined average base coverage of 155× and a BNG physical coverage of 220× was obtained (Figure 1A; Table S1). Average physical molecule lengths were 534 bp for ILL paired-end inserts, 11 kbp for ONT, 19 kbp for PB, 20 kbp for 10X, and 98 kbp for BNG optical maps (Figure 1B; Table S1).

To assess the consistency of each of the technologies with respect to representation of the sequence content of the COLO829 cancer cell line, we determined the presence of CNAs. Unfortunately, no single CNA caller was available to detect CNAs with high resolution for all technologies. Nevertheless, low-resolution CNA calling revealed a highly similar copy number profile for each of the technologies (Figure 1C), with a correlation of copy number calls in the different datasets of 0.87–0.96 (Figure S1A). Furthermore, we compared our copy number calls with those generated in previous bulk[26] and single-cell[32] sequencing of COLO829. The overall CNA landscape of the bulk sequencing and the dominant cluster from single-cell sequencing is very similar to the one we obtained (Figure S1B), with a correlation of 0.99 (bulk) and 0.97 (single cell group A) (Figure S1C). However, the previously described sub-clonal single cell clusters (B–D) possess some distinct copy number aberrations that are not observed in our bulk sequencing datasets (i.e., extra copy of chromosome 8 in group D or lack of gain in short arm of chromosome 1), in line with the proposed continuous genomic evolution of cell lines and subculture-specific nature of these events. Finally, classical FISH analysis for six genomic locations of the culture used in our study confirmed the sequencing-derived chromosomal copy number states (Figure S1D).

### Generation of a somatic structural variation consensus truth set

To build an accurate and comprehensive somatic SV truth set, we used a combinatorial analysis approach involving the four sequencing platforms (ILL, ONT, PB, and 10X). To avoid inconsistencies derived from nomenclature and classification of SVs in the different datasets, we focused on the detection of individual breakpoints rather than complex events, with a minimum event size of 30 bp. We used state-of-the-art SV calling tools appropriate for each of the sequencing datasets. Due to the lack of an existing benchmark and best-practices protocols in the somatic SV calling field, this study was oriented to the creation of a validated truth set and not to the benchmarking of somatic SV calling tools and sequencing technologies. Therefore, we chose optimal mapping and SV calling tools to the best of our knowledge (STAR Methods; Figure 2A). SV calling parameters were not optimized for highest precision but for high sensitivity to not miss any real event. As a result, individual candidate call sets for each technology resulted in highly variable lists of predicted somatic SVs, ranging from 92 breakpoint calls in ILL up to 6,412 for ONT, adding up to a total of 8,831 merged candidate somatic SV calls (Figure 2A). Only 18 of those somatic SV calls were found by all four sequencing approaches, and 125 SV calls were supported by at least two call sets (Figure S2A).

To make an initial assessment of accuracy, we selected 88 high-confidence SV candidates for PCR validation based on visual inspection of the mapped reads using Integrated Genome Viewer (IGV). In addition, we randomly selected 296 additional SV candidates for PCR validation. Based on short- and long-read sequencing of the PCR products, 63 of these breakpoints were labeled as PCR validated (Figure S2B). Moreover, we decided to perform a separate validation of all 8,831 somatic SV calls from the union of the four SV call sets, using a

## Resource

CellPress

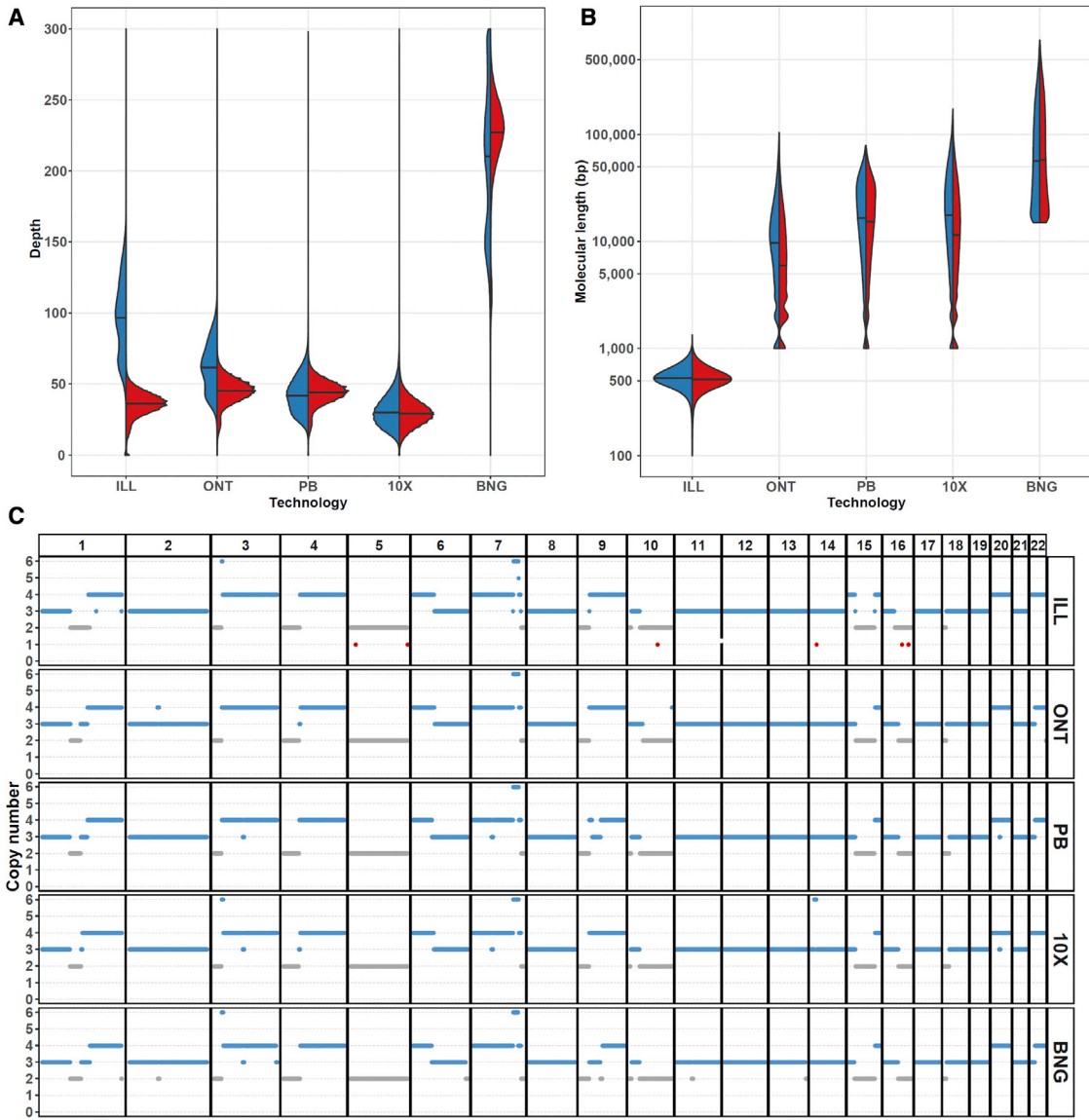

**Figure 1. Overview of the COLO829 multi-technology genomic dataset**

(A and B) Sequencing depth (A) and log-scaled molecular analysis length (B) distributions per technology dataset for COLO829 (blue) and COLO829BL (red). Means are indicated by horizontal black lines.

(C) Copy number profile of COLO829 calculated independently for each of the datasets.

capture-based enrichment method using multiple probes flanking and overlapping each candidate break junction (STAR Methods). Based on the short-read sequencing of the enriched products, 114 breakpoints were labeled as capture validated (Figure S2B). Lastly, we used the 52 BNG somatic SV calls as an additional layer of validation. In total, 137 SV calls were validated by at least one of the aforementioned methods. In addition, 78 SV calls were not validated but still supported by more than one technology (Figures 2A and S2C).

Next, we manually curated these 215 SV calls that were either validated or supported by multiple technologies. Based on visual inspection of the genomic alignment data from each of the sequencing sets and from the validation experiment results, we assessed each SV call individually. We found that 48 calls were real events but also had evidence in the germline control, and another 99 were considered false positives as the supporting or reference data were very noisy at the given genomic location (also in the independent validation data). This may reflect the impact of low-confidence regions in the reference genome, for which unambiguous mapping of sequencing reads is complicated due to simple sequence or repeat content. Taken together, we conclude that 68 of the SV candidates are real somatic events and thus considered as our truth set (Figures 2A and S2C; Table S2). To verify the efficacy of our manual curation approach, we manually curated 179 randomly selected additional SV calls that were supported by a single technology and not validated

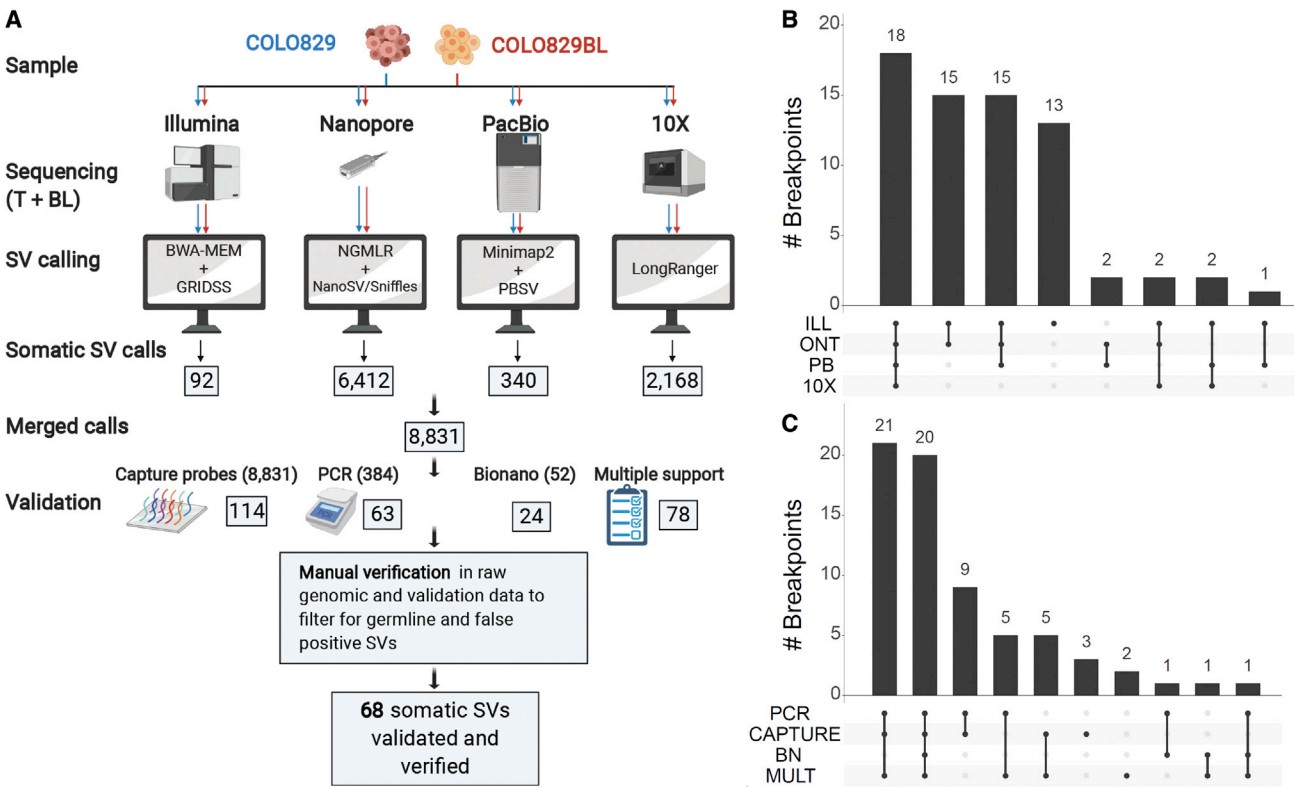

**Figure 2. Generation of a validated somatic SV truth set**

(A) State-of-the-art somatic SV calling pipelines were used independently for each technology dataset. The number of somatic SV candidates identified are indicated in boxes. Overlapping variant calls obtained by the different platforms were merged and independently validated using a combination of targeted enrichment with hybrid capture probes followed by next-generation sequencing, PCR, and Bionano genomics. Validated somatic SV candidates and calls supported by more than one dataset were manually curated, leaving a total of 68 somatic SVs in the truth set.

(B and C) Intersections between the 68 somatic SVs in the truth set and the original SV call sets (B) and the validation results (C) are shown. 10X, 10× Genomics; BN, Bionano; ILL, Illumina HiseqX; MULT, support by multiple sequencing platforms; ONT, Oxford Nanopore; PB, PacBio.

and therefore left out from the candidate SV curation pipeline. All these SV calls were either germline events (63; 35%) or false positives due to noisy mapping data (116; 65%) (Figure S2D). To corroborate that the breakpoint-merging threshold of 200 bp used in our filtering pipeline was not too stringent, we did a re-run of the filtering analysis step using 1,000 bp as a merging threshold, resulting in an extra 121 breakpoints supported by more than two technologies. We verified these breakpoints similarly as the original filtering pipeline and classified 70 as false positives and 51 as germline, resulting in no added value for the truth set (Figure S2E).

Of the compiled set of 68 validated somatic SVs in COLO829, 55 (81%) were present in more than two original call sets, including the 18 SVs detected by all technologies (Figure 2B). Moreover, most of the SVs were validated by capture-based enrichment and by PCR (50; 74%). In addition, eight somatic SVs were validated by capture-based enrichment, but not by PCR, and seven somatic SVs were validated by PCR, but not by capture-based enrichment. Of the remaining three SVs, one was validated by BNG and two were not validated by any targeted assay but are supported by multiple technologies and manually verified by inspection of raw sequencing data from both tumor and normal samples (Figure 2C). The resulting

somatic SV truth set is presented in Table S3 and freely available as a variant call format (VCF) file. We also provide a GRCh38-lifted version of the somatic SV truth set.

**Characterization of the COLO829 somatic SV truth set**

The carefully curated and validated somatic SV truth set consists of 38 deletions, 3 insertions, 7 duplications, 7 inversions, and 13 translocations (Figure 3A). Most of the deletions (24; 61%) are larger than 10 kbp, and seven are smaller than 100 bp. There are also three duplications and three inversions larger than 10 kbp. Two tumor-driver genes are affected by somatic SVs in COLO829 (Table S3). First, there are two large heterozygous deletions (72 and 141 kb) in FHIT, located in the fragile site FRA3B on chromosome 3, which is commonly affected by somatic SVs.[2] Second, there is a homozygous 12-kbp deletion affecting PTEN on chromosome 10. There are breakpoints in all chromosomes except 2, 13, 17, and 21 (Figure 3B). These chromosomes also do not show any CNA event. Annotation of the somatic SV breakpoints with gnomAD-SV,[33] segmental duplications, simple repeats, or microsatellites from the University of California, Santa Cruz (UCSC) genome browser did not reveal any overlap.

Frequently, SVs do not occur as simple isolated events but are part of a complex landscape induced in a single event like, for

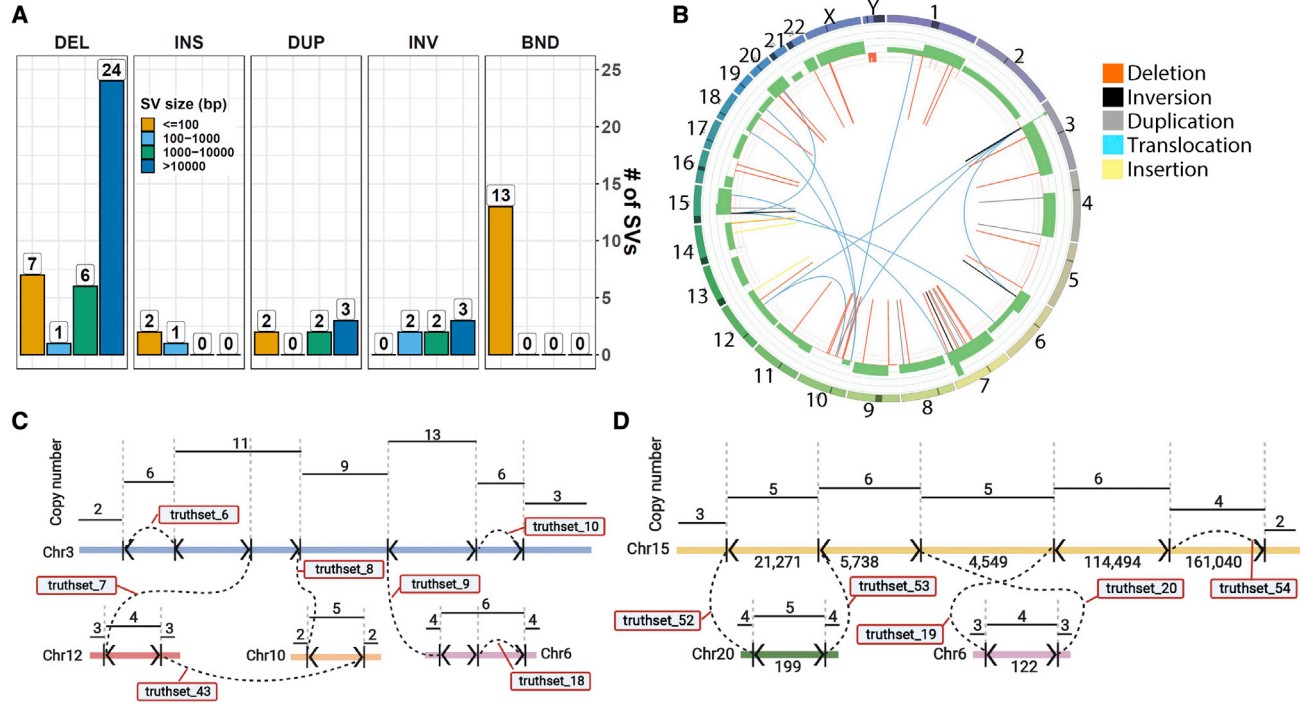

**Figure 3. Characterization of the somatic SV truth set**

(A) Distribution of different types of SVs in the COLO829 truth set, divided in size bins. Translocations (BND) are assigned a size of 0 bp.

(B) Correlation between CNAs and somatic SVs in the COLO829 truth set. The circos plot shows copy number gains (green) and losses (red) and somatic SVs. Each copy number change is expected to be flanked by an SV event. Two complex breakage-fusion-bridge events are present in COLO829.

(C) The first one occurs in chromosome 3 (blue), with templated insertions from chromosomes 6 (pink), 10 (green), and 12 (red) (see also Video S1 for an animation of the proposed mechanism shaping this event).

(D) The second one occurs in chromosome 15, with templated insertions from chromosomes 6 (pink) and 20 (green). Breakpoints are indicated by vertical lines with arrowheads showing breakpoint orientations. Dashed lines indicate junctions between two breakpoints. Break junctions are labelled with truth set SV ID number (Table S3).

---

example, chromothripsis or due to a cascade of events over time like breakage-fusion-bridge cycles. There are also two clusters of complex chained somatic SVs that affect three or more chromosomes and involve more than five breakpoint junctions. Both of them resemble breakage-fusion-bridge events, since they are flanked by foldback inversions and show oscillating copy number profiles.[2] One of them occurred in chromosome 3 and involves four foldback inversions, two of which have templated insertions from chromosomes 10 and 12 and chromosome 6, respectively (Figure 3C). The breakpoint and copy number profile of chromosome 3 can be fully explained by four cycles of breakage-fusion-bridge followed by chromatid duplication through a whole-genome doubling event. Initiated by replication of unrepaired double-stranded break, the unstable chromosome 3 (due to the presence of two centromeres in a single chromatid) underwent a further three more rounds of breakage-fusion-bridge (BFB) with a fragment of chromosome 6 inserted prior to the third doubling cycle and fragments of chromosomes 10 and 12 inserted immediately after the fourth doubling cycle. A stable state was achieved after the final breakage through repair to one of the centromeres (Video S1). The other BFB event occurred on chromosome 15 and includes templated insertions from chromosomes 6 and 20 (Figure 3D).

The donor locations of these templated insertions are not affected by SV events.

To evaluate the completeness of the somatic SV truth set, we compared it with the somatic CNA calls, since each CNA should have SV breakpoints or telomeres at either end. We found 43 total CNA breakpoints that are not telomeric ends of chromosomes. Of these, 26 (60%) are concurrent with an SV breakpoint. We evaluated the rest of the CNAs in the raw genomic data (Table S4). From these remainder CNAs, six copy number breakpoints (14%) are present in the germ line, flanking heterozygous germline CNA events that are homozygous in the tumor through a somatic loss of the other allele. The SV break junctions of these CNAs are germline and therefore not part of the truth set. Finally, there are 11 somatic CNA breakpoints (26%) not concurrent with an SV breakpoint. Five of these missing CNA breakpoints are located in a centromeric region (chromosomes 1, 4, 6, 14, and 16) and are likely due to a missing somatic SV involving the centromere, which is typically hard to fully resolve due to its repetitive nature. For another two missing CNA breakpoints (chromosome 3 and chromosome 9), breakends can be found in the raw ILL dataset, meaning an SV breakpoint was found, but the SV junction partner could not be unequivocally determined. GRIDSS2 annotation did reveal that the chromosome 3 single

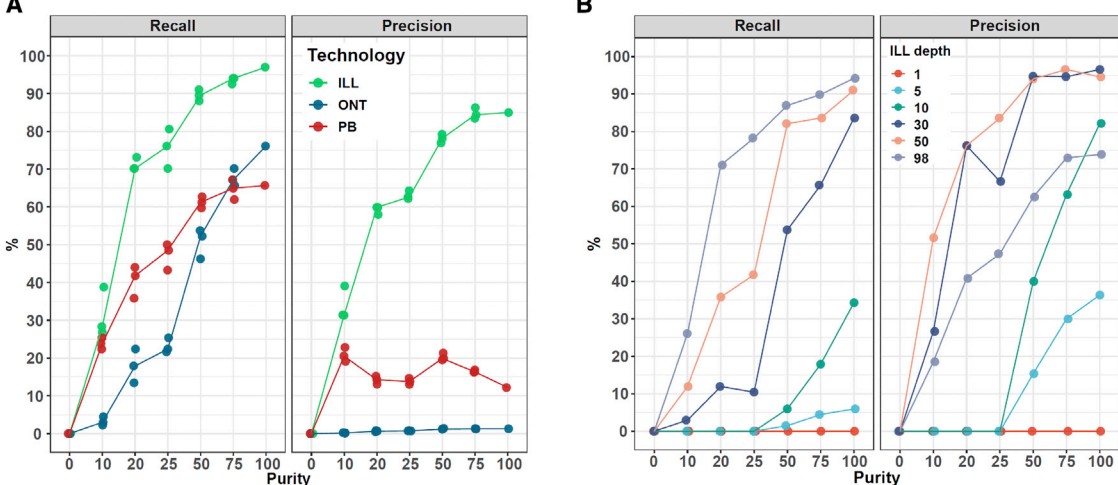

**Figure 4. Recall and precision of somatic SV calling as function of tumor purity and sequencing depth effect**
Different tumor purities (0%, 10%, 20%, 25%, 50%, 75%, and 100%) were simulated by mixing data from COLO829 and COLO829BL for the ILL, ONT, and PB datasets.
(A) Somatic SV calling was performed independently for each purity subset, and recall (left) and precision (right) were calculated against the COLO829 somatic SV truth set. Lines represent the median of independent triplicate measurements.
(B) For each tumor purity subset in the ILL dataset, different sequencing depths (1×, 5×, 10×, 30×, 50×, and 98×) were sampled. Somatic SV calling was performed independently for each sequencing depth and tumor purity subset, and recall (left) and precision (right) were calculated against the COLO829 somatic SV truth set.

break does map to one of the centromeres. Four more missing CNA breakpoints flank two supposed deletions in chromosome 1, but no SV call in these locations can be found for either COLO829 or COLO829BL in any of the datasets. Manual inspection of the raw data for these CNAs (Figures S3A and S3B) indicates that these CNAs may actually reflect heterozygous germline events followed by loss of heterozygosity (LOH) as witnessed by the lower read coverage in the COLO829BL as compared with the flanking segments. Furthermore, one CNA involves a long interspersed nuclear element (LINE)-rich region, while the other overlaps with a segmental duplication.

Next, we compared our somatic SV truth set with the somatic SV calls presented by Arora et al.[26] They provide two different somatic SV call sets, one generated by the HiSeq platform with 77 somatic SV calls and the other by the NovaSeq platform with 75 somatic SV calls. Since these were provided based on GRCh38 genomic coordinates, we lifted our somatic SV coordinates over to GRCh38. We found that 58 (75.34%) and 59 (78.6%) of the somatic SV calls for the HiSeq and the Novaseq call sets, respectively, overlapped with our somatic SV truth set on both sides of the SV (Figure S3). We manually inspected the 20 non-overlapping somatic SV calls from the Arora et al. dataset in our raw ILL, ONT, and PB data (Table S5). In the long-read raw data (ONT and PB) only 3 out of the 20 have some support (maximum three reads). In the ILL raw data, 9 out of the 20 have limited evidence, with only one or a few supporting reads. Only four of these nine SV calls passed bioinformatic calling criteria in our original ILL somatic SV calls, but none of these were called by any other technology or independently validated by more sensitive PCR or targeted capture and deep sequencing. Therefore, we considered these candidates as

technology-specific noise and discarded them from our truth set, although we cannot formally exclude that these are real variants that are present at very low frequency (<1% in the sample). Finally, 13 SVs are present in our truth set and not in the Arora et al. dataset. All were detected by at least two different sequencing techniques and independently validated.

### Effect of tumor purity and sequencing depth on somatic SV calling

To demonstrate the utility of the COLO829 somatic SV truth set, we evaluated the effect of tumor purity, which is highly variable among clinical samples, on SV calling. We used the available raw datasets and simulated tumor purities of 75% (TP75), 50% (TP50), 25% (TP25), 20% (TP20), and 10% (TP10) by random *in silico* mixing of the genomic data from COLO829 and COLO829BL for ILL, ONT, and PB, respectively. We performed SV calling independently on each of these mixed sets and on the original tumor file (100% purity [TP100]) and the normal file (0% purity [TP0]). We then calculated the recall (percentage of truth set found) and precision (percentage of calls that belong to the truth set). With the standard settings used, somatic SV recall and precision were found to be highly dependent on tumor purity for all three technologies (Figure 4A). At TP75 and TP100, recall is the highest, with >94% (ILL), >67% (ONT), and >65% (PB). With TP50, the recall slightly decreases to 90% (ILL), 52% (ONT), and 61% (PB). For purities lower than TP50, the recall decreases further to <76% (ILL), <22% (ONT), and <48% (PB). Precision follows a similar trend in the case of ILL, with precisions >78% for purities larger than TP50 and a drop to 63% in TP25. In the case of ONT and PB, the higher number of false positives severely impact the precision rates, potentially reflecting

the maturity level of platform-specific tools for somatic SV detection in tumor-normal paired samples but also presenting opportunities for further analysis parameter and tool optimization.

Sequencing depth is another essential parameter to consider in tumor-sequencing projects, as it represents a trade-off decision between variant detection sensitivity and costs. To investigate the effect of sequencing depth in combination with tumor purity in somatic SV detection, we took one of the triplicates from each of the simulated ILL tumor purities (98× coverage) and subsampled them to 50×, 30×, 10×, 5×, and 1× depths. We again performed somatic SV calling using the same standard pipeline on each of these simulated sets and calculated recall and precision (Figure 4B). We observed that, for depths of 50× and 98× and tumor purities over 50%, recall was over 82%. In the case of 98×, even at TP20, a recall of 71% was obtained, whereas for 50×, at TP25, the recall decreased to 42%. For 30× sequencing depth, at TP100, recall was 84%, but at TP50, there was a decrease to 54% and, at TP25, further to 10%. For lower coverages, recall was low. Surprisingly, depths of 30× and 50× had a higher precision at all tumor purities than 98×, with precision around 95% over TP50, compared with approximately 70% for 98×. While this could in theory be explained by the presence of subclonal SVs that are not included in the truth set but become detectable at higher sequencing depth, this might also be caused by stochastic effects due to increased measurement noise at higher sequencing depth, which increases the number of false positives and therefore reduces precision (although recall is not affected). Another possible explanation is that SV-detection tools have been developed and optimized using sequencing depths around 30× and therefore function better at those depths, needing optimized parameters for optimal performance at different sequencing depths. Further optimization of analysis tools and settings and deeper sequencing may resolve these issues.

### Benchmarking against the COLO829 truth set

To aid future benchmarking using the COLO829 truth set described, we developed a script to directly compare SVs with this or other future benchmarks. This script compares SVs at the breakpoint resolution to generate a precision and recall plot. To demonstrate its use, we compared the ILL, ONT, and PB calls used for the development of the truth set with the most updated versions of SV variant calling tools available at the time of submission of this work (Figure S4). We observed an improvement in recall for the updated version of PBSV. The drop in precision for GRIDSS is likely due to differential manual pre-processing of the original GRIDSS file, which was substituted by automatic filtering in the updated version. Surprisingly, a drop in recall can be observed in the updated version of Sniffles, while maintaining the low precision. We did not further analyze in detail the causes for these changes, as this is beyond the scope of the current work, but the framework presented does allow for a versatile approach to bioinformatic tool and parameter optimization. We included the updated VCF files from each technology in the updated data bundle. Any other benchmarking with own VCF files can be performed using the R script with our COLO829 or other future truth sets.

## DISCUSSION

We produced a carefully curated and validated somatic SV truth set by building upon the strengths of different sequencing technologies. Bioinformatic integration of results and large-scale independent validation strategies turned out to be a powerful approach for reducing the large number of candidate events obtained. Manual curation and inspection of raw sequencing data were, however, essential to exclude sequencing or mapping artefacts and remaining germline events. These somatic false positives are thus germline false negatives and were likely included in the initial somatic SV calls due to the lower sequencing analysis depths for the control sample as compared with the tumor (typically 3-fold lower) in combination with specific local genomic characteristics (e.g., lower average coverage due to, for example, local GC content or involving low-complexity sequences).[34]

While reconstruction of the derived chromosomal tumor genome topology based on the 68-truth-set somatic SVs results in an overall stable genomic configuration for most derived chromatids harboring a single centromere and two telomeres, some breakpoint junctions are still clearly missing. This is corroborated by the fact that not for all CNAs breakpoint junctions were identified at either end. Our results indicate that these missing events typically involve centromeric regions that are not directly accessible by any current sequencing technology. Annotation data provided by the GRIDSS2 SV caller[35] suggest a junction between a single break-end in chromosome 3 and the centromere in chromosome 1, which shows a copy number change. Probably, this cannot be resolved directly due to the repeated nature of the centromeric region. When excluding the missing events that likely involve centromeres, there are two copy number aberrations that remain unexplained by the truth set, providing room for further improvement based on the existing or to-be-generated data. Of course, we can formally not exclude that there are more events missing from our truth set due to limitations in current sequencing and data analysis approaches, for example, due to inaccessibility of centromeres, telomeres, or other repetitive elements. Therefore, we do recommend following up promising novel candidates that emerge in future benchmarking studies, with orthogonal validations to further improve the current truth set.

This study was not designed to compare performance of sequencing platforms or data analysis pipelines, since that benchmarking would require the very latest platform, chemistries, and pipeline versions to be useful. Nevertheless, some interesting observations can be made. First, there is clear complementarity between the various platforms for the comprehensive identification of all real events. However, bioinformatic pipelines for somatic SV detection are still clearly in different stages for the different platforms, with the most commonly used Illumina-based approaches yielding lowest numbers of false positives. For example, joint calling in the tumor and normal sequencing data and cancer-specific somatic filtering are very important to reduce false-positive rates in somatic SV calling. However, such an approach was only available for the Illumina dataset, as no somatic-specific callers or protocols exist for the other datasets yet. We believe future tool optimization for somatic SV calling, assisted by truth variant call datasets as well as

the development of platform-specific germline and artefact-filtering datasets ("pools of normals") based on large numbers of samples, will effectively address this challenge. Second, data analysis pipelines yield different annotations for the same event. This calls for further standardization of variant annotation and nomenclature, although some observed differences are intrinsic to the use of short- and long-read technologies. For example, a long-templated insertion may be called as two independent translocations by short-read SV callers, while long-read-based technology would detect this readily as an insertion. Third, despite previous studies showing the added value of long reads for SV detection for germline events, our somatic SV truth set is resolved almost in its entirety with the ILL short-read dataset. Likely, this is mostly due to the more advanced somatic SV calling pipelines developed for short-read data than for long-read data, as previously discussed. However, this observation may also be explained by fundamental differences between germline and somatic SVs, such as the overall distribution throughout the genome, the involvement of repetitive regions, and the total number of such events. With further methodological advances in the somatic and germline SV calling, these differences will undoubtedly be further characterized and better understood.

Apart from the benchmarking opportunities provided by our truth set, the COLO829 cell line has the advantage that it is, in contrast to real tumor samples, a renewable source. Therefore, it can be used for assessing the impact of future platform developments or the performance of completely new technologies for somatic mutation detection by generating new datasets from the same cell line. However, although the COLO829 cell line is representative of SV as observed in cancer, including small and large CNAs (including aneuploidies) and both simple and complex SV events, it is not necessarily representative in all aspects for real tumor samples. First, tumor samples do typically not consist of tumor cells only but are a mix of tumor and normal cells (e.g., stromal cells and infiltrating immune cells). We show that the raw data obtained in this study can be used effectively to mimic variable tumor purity and that the truth set is instrumental for assessing the performance of the bioinformatic data analysis tools at variable tumor purity. As expected, our results show that both recall and precision heavily depend on tumor purity for all platforms. Secondly, tumors evolve continuously and are typically genetically heterogeneous, especially primary tumors, involving potentially subclonal SV events. While the COLO829 cell line is relatively stable at the genomic level, it has a certain level of genetic heterogeneity and is subject to mutation accumulation and evolution throughout culture like any cell line. This variation is dynamic and might differ between cell line isolates, as already demonstrated by the various studies on this cell line,[31,32] and thus limit the utility of a single defined truth set obtained as presented here. Therefore, novel somatic SVs not present in our current truth set should be validated independently, especially when data were generated using different batches of cells from the COLO829 cell lines. Finally, tumors are in general very heterogeneous both within the context of a specific tumor type but especially between tumor types. For example, microsatellite instable (MSI) tumors show a high number of small indels,[36] homologous-recombination-deficient

(HRD) tumors present many deletions with microhomology and large duplications,[37] and pediatric hematological cancers usually show very low mutational load but enhanced levels of somatic SVs, although often involving specific but complex genomic loci (e.g., the immunoglobulin H [IgH] locus).[38,39] The specificity for capturing such heterogeneity effectively or the impact of specific genomic events that may co-occur in a given tumor sample, like, for example, whole-genome duplication or chromothripsis, on overall performance of a specific sequencing technique or data analysis tool is of course not captured in a single cell line and requires the development of complementary datasets. The COLO829 truth set should therefore be used with caution, and analyzing additional cancer cell lines with matching normal cell lines may provide an attractive route for future improvements, as these represent, in principle, an endless source of genomic material for benchmarking of future DNA analysis technologies but also for quality monitoring in routine production labs under accreditation. However, availability of suited cell lines that represent the full genetic diversity of cancer is a clear limitation. Ideally, one would thus resort to thoroughly analyzed real tumor samples, even though, in practice, availability of sufficient material for multi-lab and multi-technology analyses can be problematic and sharing and reusing of patient material and data may require complex consenting and legal procedures. The use of synthetic samples could also be a complementary approach,[40] although its utility for mimicking complex structural variation remains to be demonstrated and technical challenges may arise when the sequencing technology that one wants to benchmark requires input of high-molecular-weight molecules.

Taken together, we believe the SV truth set described here as well as the underlying raw data are valuable resources for benchmarking and fine-tuning analysis settings of somatic SV calling tools, but the data may also be used for the development of novel analysis tools, for example, phasing of structural variants. All analysis results and raw data are publicly available to enable such applications without access restrictions (ENA: PRJEB27698; an overview of the available data and specific access link can be found at Table S6). We demonstrate this utility by analyzing the impact of tumor purity and sequencing depth on SV recall and precision for different technologies, thereby providing valuable insights in the potential impact of technology platform choice and experimental design in relation to diagnostic accuracy and overall costs. Furthermore, these results highlight the need of benchmarking somatic SV detection methods at different tumor purities and sequencing depths rather than under a single fixed condition, since these parameters are highly variable within and between cohorts and can result in strong performance variation.

### Limitations of the study
In this study, we used different sequencing techniques and analyzed the tumor cell line genome to a variable but limited depth. As a consequence, subclonal events with a frequency below 5% are likely missed in our analyses. This effect could potentially be larger for events that can only be detected with a specific technology and for which sequencing coverage was in the range of only 30× to 50× (PB and 10X). In addition, we

sequenced tumor samples to a higher depth than the germline control samples, which is a routine practice to compensate for variable tumor purity and heterogeneity, but it has been shown that performance of somatic variant calling is affected by unbalanced coverage of test and reference samples.[34] Nevertheless, it needs to be demonstrated whether this effect is also present for structural variant calling and the variant calling tools used. Finally, it should be noted that the structural variant call set was manually curated. Although all true events were independently validated, curation may have incorrectly removed real variants.

## STAR★METHODS

Detailed methods are provided in the online version of this paper and include the following:

- KEY RESOURCES TABLE
- RESOURCE AVAILABILITY
  - Lead contact
  - Materials availability
  - Data and code availability
- EXPERIMENTAL MODELS AND SUBJECT DETAILS
- METHODS DETAILS
  - Genomic analyses per technology
  - Depth and molecular length calculations
  - Copy number analysis
  - Validations
  - SV selection pipeline
  - Liftover to GRCh38
  - Comparison to external sources
  - Tumor purity and sequencing depth analysis
  - Benchmarking against the COLO829 truth set
- QUANTIFICATION AND STATISTICAL ANALYSIS

### SUPPLEMENTAL INFORMATION

### ACKNOWLEDGMENTS

We thank Pacific Biosciences and BioNano for their kind support generating and analyzing data. J.E.V.-I. is supported by the Gieskes Strijbis Foundation (1816199). This work was performed as part of the EU-funded Horizon2020 EUCANcan project (funding to E.C.) and the Netherlands X-omics Initiative funded by NWO, project 184.034.019.

### AUTHOR CONTRIBUTIONS

Conceptualization, J.E.V.-I., B.Y., R.J.A.F., W.P.K., and E.C.; investigation, J.E.V.-I., N.J.M.B., E.d.B., M.N., and I.R.; formal analysis, J.E.V.-I., D.L.C., J.E., J.K., S.v.L., T.M., P.P., M.J.v.R., and A.M.W.; validation, N.J.M.B., I.R., M.G.M.R., and M.J.v.R.; visualization, J.E.V.-I.; writing – original draft, J.E.V.-I., W.P.K., and E.C.; writing – review and editing, J.E.V.-I. and E.C.; supervision, W.P.K. and E.C.; funding acquisition, E.C. and W.P.K.

### DECLARATION OF INTERESTS

A.M.W. is an employee and shareholder of Pacific Biosciences. W.P.K. is an employee and shareholder of Cyclomics B.V.

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

**Cell Genomics**
**Resource**

## STAR★METHODS

### KEY RESOURCES TABLE

| REAGENT or RESOURCE | SOURCE | IDENTIFIER |
|---|---|---|
| **Critical commercial assays** | | |
| Truseq Nano reagent kit | Illumina | Cat#:20015965 |
| Hiseq X Ten reagent kits | Illumina | V2.5 |
| MinION/GridION flow cells | Oxford Nanopore Technologies | R9.4 |
| Sequel chemistry | Pacific Biosciences | V5.0 |
| Sequel binding kit | Pacific Biosciences | Cat#:101-365-900 |
| Sequel sequencing kit | Pacific Biosciences | Cat#:101-309-500 |
| Chromium Genome platform | 10X genomics | N/A |
| Novaseq platform | Illumina | V1 |
| SP Blood & Cell culture Isolation chip | Bionano genomics | Cat#:80030 |
| Direct Label and Stain kit | Bionano genomics | Cat#:80005 |
| Saphyr chip and instrument | Bionano genomics | Cat#:20367 |
| Biotin-labelled custom targeted probes | Twist Biosciences | Cat#:100253; 100255; 100527; 100400 |
| SpectrumOrange probes | Abbott Vysis | Cat#:08N31-030 |
| Vysis ALK Break Apart FISH Probe Kit | Abbott Vysis | Cat#:06N38-023 |
| SpectrumAqua probes | Leica Biosystems | N/A |
| Leica DM5500 fluorescence microscope | Leica Biosystems | N/A |
| **Deposited data** | | |
| Raw and mapped genomic data for COLO829 and COLO829BL | This paper | ENA: PRJEB27698 |
| Raw and processed somatic SV VCFs and CNA calls | This paper | Zenodo: http://doi.org/10.5281/zenodo.4716169 |
| Somatic SV and CNA calls | Arora et al., 2019 | https://www.nygenome.org/bioinformatics/3-cancer-cell-lines-on-2-sequencers/ |
| **Experimental models: Cell lines** | | |
| COLO829 | ATCC | CRL-1974 |
| COLO829BL | ATCC | CRL-1980 |
| **Software and algorithms** | | |
| BWA mem v0.7.5 | Li, 2013 | https://github.com/lh3/bwa |
| GATK v.3.4-46 | DePristo et al., 2011 | https://gatk.broadinstitute.org/hc |
| GRIDSS v.2.0.1 | Cameron et al., 2021 | https://github.com/PapenfussLab/gridss |
| NGMLR v.0.2.6 | Sedlazeck et al., 2018 | https://github.com/philres/ngmlr |
| Sniffles v.1.0.9 | Sedlazeck et al., 2018 | https://github.com/fritzsedlazeck/Sniffles |
| NanoSV v.1.2.2 | Cretu Stancu et al., 2017 | https://github.com/mroosmalen/nanosv |
| SURVIVOR v.1.0.6 | Jeffares et al., 2017 | https://github.com/fritzsedlazeck/SURVIVOR |
| Minimap2 v.2.11-r797 | Li, 2018 | https://github.com/lh3/minimap2 |
| pbsv v.2.0.1 | | https://github.com/PacificBiosciences/pbsv |
| LongRanger-WGS v2.2.2 | | https://support.10xgenomics.com/genome-exome/software/pipelines/latest/using/wgs |
| Bionano Access | | https://bionanogenomics.com/support-page/bionano-access-software/ |
| Sambamba v.0.6.5 | Tarasov et al., 2015 | https://github.com/biod/sambamba |
| Bedtools v.2.25.0 | Quinlan and Hall, 2010 | https://bedtools.readthedocs.io/en/latest/ |
| Picard v.1.141 | | http://broadinstitute.github.io/picard |

*(Continued on next page)*

*Continued*

| REAGENT or RESOURCE | SOURCE | IDENTIFIER |
|---|---|---|
| BIC-SEQ2 v0.7.2 | Xi et al., 2016 | compbio.med.harvard.edu/BIC-seq/ |
| Ginkgo | Garvin et al., 2015 | http://qb.cshl.edu/ginkgo |
| Manta v.0.29.5 | Chen et al., 2016 | https://github.com/Illumina/manta |
| Primer3 v.1.1.4 | Untergasser et al., 2012 | https://github.com/primer3-org/primer3 |
| SV-plaudit | Belyeu et al., 2018 | https://github.com/jbelyeu/SV-plaudit |
| Integrated Genome Viewer (IGV, v.2.4.0) | Robinson et al., 2017 | https://software.broadinstitute.org/software/igv/ |
| ENSEMBL Assembly converter | | https://www.ensembl.org/Homo_sapiens/Tools/AssemblyConverter |
| StructuralVariantAnnotation v1.2.0 | Cameron and Dong 2019 | https://www.bioconductor.org/packages/release/bioc/html/StructuralVariantAnnotation.html |
| BioRender | | https://biorender.com/ |
| Other | | |
| All code used in the analysis | This paper | https://github.com/UMCUGenetics/COLO829_somaticSV; https://doi.org/10.5281/zenodo.6426985 |
| Code used for tumor purity and sequencing depth analysis | This paper | https://github.com/UMCUGenetics/tumps; https://doi.org/10.5281/zenodo.6426991 |

## RESOURCE AVAILABILITY

### Lead contact
Further information and requests for resources should be directed to and will be fulfilled by the lead contact, Edwin Cuppen (e.cuppen@hartwigmedicalfoundation.nl).

### Materials availability
This study did not generate new unique reagents or materials.

### Data and code availability
- All genomic data generated and used in this study are available in ENA with project ID ENA: PRJEB27698. Raw, somatic and truth set VCF files, including updated calls used in benchmarking and GRCh38 lifted-over truth set, and CNA files are available at Zenodo: http://doi.org/10.5281/zenodo.4716169. More data availability details are available in Table S6.
- All code used in the preparation of the somatic SV truth set is available at GitHub: https://github.com/UMCUGenetics/COLO829_somaticSV (https://doi.org/10.5281/zenodo.6426985). The code used for simulations of tumor purity and sequencing depth is available at GitHub: https://github.com/UMCUGenetics/tumps (https://doi.org/10.5281/zenodo.6426991).
- Any additional information required to reanalyze the data reported in this paper is available from the lead contact upon request.

## EXPERIMENTAL MODELS AND SUBJECT DETAILS

COLO829 (ATCC® CRL-1974™) and COLO829BL (ATCC® CRL-1980™) cell lines were obtained from ATCC in September 2017. A single batch of cells was thawed and cells were expanded and grown according to standard procedures as recommended by ATCC. Cell pellets were split for technology-specific DNA isolation at 33 days (COLO829 & COLO829BL for the ILL and ONT datasets), 35 days (COLO829 for the PB, 10X and BNG datasets) and 23 days (COLO829BL for the PB, 10X and BNG datasets).

## METHODS DETAILS

### Genomic analyses per technology
#### Illumina
COLO829 and COLO829BL libraries were prepped with Truseq Nano reagent kit and sequenced on the HiSeq X Ten platform using standard settings and reagent kits (chemistry version V2.5). Reads were mapped to GRCh37 with BWA mem (version 0.7.5,[41]), followed by indel realignment with GATK (v3.4-46,[42]). SVs were called jointly for COLO829 and COLO829BL with GRIDSS (v2.0.1,[43]). Somatics SVs were filtered with the GRIDSS somatic SV filtering script (https://github.com/PapenfussLab/gridss/blob/master/scripts/gridss_somatic_filter.R).

### Nanopore

COLO829 and COLO829BL libraries were sequenced on the MinION and GridION platforms using R9.4 flow cells. Reads were mapped to GRCh37 with NGMLR (v0.2.6, default parameters,[44]) with default parameters. SV calling was performed with both NanoSV (v. 1.2.2, default parameters,[20]) and Sniffles (v1.0.9, parameters "–report_BND –genotype",[44]) for COLO829 and COLO829BL separately. All SV calls for both NanoSV and Sniffles were merged with SURVIVOR (v1.0.6,[45]) with a distance of 200 bp and calls with evidence in COLO829BL for NanoSV or Sniffles were discarded.

### PacBio

COLO829 and COLO829BL libraries were sequenced on the Sequel System with the 5.0 chemistry (binding kit 101-365-900; sequencing kit 101-309-500). Reads were mapped to GRCh37 with minimap2 (v2.11-r797,[46]). SVs were called jointly for COLO829 and COLO829BL with pbsv (v2.0.1, https://github.com/pacificbiosciences/pbsv/) using default parameters. Somatic SV calls were filtered by removing any call with a supporting read in COLO829BL.

### 10X

COLO829 and COLO829BL 10× genomics libraries were prepared on the Chromium platform and sequenced on the NovaSeq platform (chemistry version V1). Reads were analyzed with the LongRanger WGS pipeline (v2.2.2) separately for COLO829 (somatic mode) and COLO829BL (default parameters). SV calls for COLO829 and COLO829BL were merged with SURVIVOR (v. 1.0.6,[45]) with an overlap distance of 200 bp and SV calls with evidence in COLO829BL were discarded.

### Bionano

DNA for COLO829 and COLO829BL was labelled using the Bionano Direct Label and Stain (DLS) kit. The labelled DNA was linearized in a Saphyr chip and imaging was performed on the Saphyr instrument. SV calling was performed on the Bionano Access platform. For each sample, 1.5 million cultured cells were used to purify ultra-high molecular weight DNA using the SP Blood & Cell Culture DNA Isolation Kit following manufacturer instructions (Bionano genomics, San Diego USA). Briefly, after counting, white blood cells were pelleted (2200 g for 2 mn) and treated with LBB lysis buffer and proteinase K to release genomic DNA (gDNA). After inactivation of proteinase K by PMSF treatment, genomic DNA was bound to a paramagnetic disk, washed and eluted in an appropriate buffer. Ultra-High molecular weight DNA was left to homogenize at room temperature overnight. The next day, DNA molecules were labeled using the DLS (Direct Label and Stain) DNA Labeling Kit (Bionano genomics, San Diego USA). Seven hundred and fifty nanograms of gDNA were labelled in presence of Direct Label Enzyme (DLE-1) and DL-green fluorophores. After clean-up of the excess of DL-Green fluorophores and rapid digestion of the remaining DLE-1 enzyme by proteinase K, DNA backbone was counterstained overnight before quantitation and visualization on a Saphyr instrument. A volume of 8.5 μL of labelled gDNA solution of concentration between 4 and 12 ng/ul was loaded on the Saphyr chip and scanned on the Saphyr instrument (Bionano genomics, San Diego USA). A total of 1.6 Tb and 1.5 Tb of data was collected for the cancer and blood sample, respectively.

De novo assembly Pipeline and Copy number variants calling were performed and against the Genome Reference Consortium Human Build 37 (GRCh37) HG19 human genome assembly (RefAligner version 7520). Events detected by the de novo assembly pipeline were subsequently compared against the matched blood control, and those that are absent in the assembly or the molecules of the control were considered as somatic variants (https://bionanogenomics.com/wp-content/uploads/2018/04/30190-Bionano-Solve-Theory-of-Operation-Variant-Annotation-Pipeline.pdf).

### Consolidation of SV calls

Somatic SV calls for each dataset (ILL, ONT, PB and 10X) were merged using SURVIVOR (v. 1.0.6 [45] with an overlap distance of 200 bp. All analyses aforementioned were performed genome-wide and no genomic regions from GRCh37 were filtered or ignored. In all cases a minimum SV size of 30 bp was established. Microsatellite instability falls in the indel size category and thus is not considered in this study.

## Depth and molecular length calculations

Average base depth and depth distribution for ILL, ONT, PB and 10X was calculated based on 1,000,000 random positions on the genome with Sambamba (v0.6.5,[47]). Average base depth for BNG was calculated based on the same 1,000,000 random positions using Bedtools (v2.25.0,[48]).

Average molecular length and length distribution was calculated based on insert size for ILL, read length for ONT and PB, on synthetic molecular length based on the MI tag for 10X, on optical map length for BNG. For ILL, average insert size was calculated using Picard (v1.141, http://broadinstitute.github.io/picard).

## Copy number analysis

Somatic CNA calling was performed on the ILL dataset with BIC-SEQ2 with default parameters (v0.7.2,[49]). For the remaining datasets, since no specific genome-wide CNA calling algorithms were available for each technology, BAM and optical map (xmap) files were converted to BED format using Bedtools (v2.25.0,[48]) and CNA calling was performed with Ginkgo.[50] CNA calls from the different datasets were merged using 1MB bins to calculate Pearson's correlation between datasets and for plotting.

## Validations

### Capture

For each break-junction of the merged somatic SV calls 2 capture probes of 100 bp in length were designed, one at either side of the breakpoint, with a maximum distance of 100 bp from the breakpoint at GC percentage as close as possible to 50%, for a total of

18148 custom probes. These custom capture probes were then ordered from Twist Biosciences. Then, libraries for COLO829 and COLO829BL were prepared and hybridized with the biotin-labelled custom targeted probes following the manufacturer's protocol (Twist Biosciences catalog IDs: 100253, 100255, 100527, 100400). Using streptavidin beads the hybridized DNA was pulled from the DNA pool, and amplified by PCR. Enriched targeted libraries were sequenced on the Illumina NextSeq platform. NextSeq-Capture validation sequencing data were mapped with BWA mem (v0.7.5,[41]) and SV calling was performed with Manta (v0.29.5,[51] independently for COLO829 and COLO829BL. SV calls for COLO829 and COLO829BL were merged using SURVIVOR (v1.0.6, overlap distance of 50 bp,[45]) and only calls with no evidence in COLO829BL were considered as somatic and validated.

### PCR
We selected 88 high-confidence SV candidates for PCR validation based on an initial screening of the somatic SV truth set with IGV and added 296 randomly selected additional SV candidates for a total of 384 assays. We automatically designed primers for these SV breakpoints using Primer3 (v1.1.4,[52]). PCR assays were performed on COLO829 and COLO829BL genomic DNA. Libraries were prepared for PCR results and sequenced on both the MiSeq and ONT-MinION platforms. MiSeq-PCR validation sequencing data were mapped with BWA mem (v0.7.5,[41]) and SV calling was performed with Manta (v0.29.5,[51]), independently for COLO829 and COLO829BL. ONT PCR validation sequencing data were mapped with minimap2 (v2.15,[46]), and SV calling was performed with NanoSV (v1.2.2, default parameters,[20]) independently for COLO829 and COLO829BL. Moreover, 70 additional SV calls that were shown as somatic in the Capture validation set were also subjected to PCR and products were sequenced on the MinION through the same protocol described above.

SV calls for COLO829 and COLO829BL from the Miseq-PCR and the two Nanopore-PCR sets were merged using SURVIVOR (v1.0.6, overlap distance of 50 bp,[45]). The threshold to merge SVs was tighter in the validation dataset than in the raw genomic dataset due to the highly targeted approach and the small size of a few base pairs of the amplicons. Only SV calls with no evidence in any of the COLO829BL sets were considered somatic and validated.

### FISH
For FISH validation, we selected probes that bind to 6 genomic regions, including Chromosome Enumeration Probes (CEP) for the centromeric region of chromosome 13, 16 and 18 (CEP13, CEP16, CEP18), labeled with SpectrumOrange (Abbott Vysis, Downers Grove, IL) and centromeric region of chromosome 9 (CEP9), labeled with SpectrumAqua (Leica Biosystems, Amsterdam). Furthermore, locus specific break-apart probes for chromosome 2p23 fusion (SpectrumOrange/SpectrumGreen, Vysis ALK Break Apart, Abbott Vysis, Downers Grove, IL) and 9p24 fusion (SpectrumOrange/SpectrumGreen Leica Biosystems, Amsterdam) were used.

COLO829 cells were dissociated using trypsin, counted, washed and diluted to contain a total of 50,000 cells in 100 μL. Monolayer cell suspensions were concentrated on a microscope slide using cytospin. Then, FISH was performed according to diagnostic standards. Slides were visualized on a Leica DM5500 fluorescence microscope and for each probe, 100 cells/slide were recorded.

### SV selection pipeline
Merged somatic SV calls, with an arbitrary size cut-off of 25 bp, were overlapped with the validation outcomes with SURVIVOR (v. 1.0.6,[45]) using an overlap distance of 50 bp (PCR, CAPTURE) and 1 kbp (BNG). Only somatic SV calls with support from multiple datasets and calls with support from a single dataset which were validated were selected. SVs involving unstable microsatellites were not considered as part of our analyses. All calls were manually curated by using the SV-plaudit cloud-based framework[53] that uses Samplot to generate images from SV coordinates and BAM files. We generated such images for the somatic SV calls for each dataset (ILL, ONT, PB and 10X) and for the validations (PCR-ONT, PCR-MISEQ and CAPTURE). We evaluated each of these image datasets independently and classified each somatic SV call as "somatic", "germline" or "false positive". We also used the Integrated Genome Viewer (IGV, v2.4.0,[54]) to verify some SVs. We performed the same analysis on 176 randomly selected SV calls belonging to a single dataset and which were not validated. Finally, we gathered the somatic SV calls and generated the final somatic VCF file.

### Liftover to GRCh38
Somatic SV breakpoint positions were lifted over to GRCh38 genomic coordinates using the ENSEMBL Assembly Converter (https://www.ensembl.org/Homo_sapiens/Tools/AssemblyConverter). We used a custom script in order to change the ALT field in the VCF. The lifted-over VCF file is available in the data bundle.

### Comparison to external sources
CNA calls from[26] were downloaded (HiSeq dataset, https://www.nygenome.org/bioinformatics/3-cancer-cell-lines-on-2-sequencers/) and lifted to GRCh37 genomic coordinates with liftOver (UCSC). CNA calls from the four different single cell clusters were obtained from.[32] These datasets were then merged using 1MB bins to calculate Pearson's correlation between datasets and for plotting.

The two somatic SV sets from[26] (HiSeq and NovaSeq sets, https://www.nygenome.org/bioinformatics/3-cancer-cell-lines-on-2-sequencers/) were downloaded. Since these are BEDPE files based on GRCh38 genomic coordinates, we converted our somatic SV truth set to BEDPE format and lifted it to those coordinates using the liftOver tool from UCSC. We then intersected those SV sets with our truth set using Bedtools (v2.25.0,[48]) and differentiated between SVs with overlap on both sides, overlap only on one side and not overlapping. We lifted all SVs with no overlap or one-sided overlap and manually evaluated them in our data using IGV v2.4.0,[54]).

### Tumor purity and sequencing depth analysis

For tumor purity simulations in each of the ILL, ONT and PB datasets, COLO829 and COLO829BL BAM files were randomly sub-sampled and mixed in different ratios, dependent on the sequencing depth to achieve *in silico* tumor purities of 10, 20, 25, 50 and 75 with Sambamba (v0.6.5,[47]). The same somatic SV calling pipeline used for the different datasets was applied to each of the tumor purity subsets. The resulting somatic SV file of each tumor purity subset was overlapped using a window of 100 bp with the truth set VCF to determine the number of true and false positives and true negatives. This experiment was performed in triplicate for each tumor purity and each technology with the original COLO829 BAM file as positive control (100% tumor purity) and the original COLO829BL BAM file as negative control (0% tumor purity).

For sequencing depth simulations using the ILL dataset, one of the triplicates from each tumor purity simulation was selected together with the COLO829 and COLO829BL files. Each of these BAM files was subsampled to depths of 1×, 5×, 10×, 30× and 50× (plus the original 98×) with Sambamba (v0.6.5,[47]). Somatic SV calling was performed independently for each of the subsets and the resulting somatic SV VCF file was overlapped with the truth set to determine the number of true and false positives and false negatives.

### Benchmarking against the COLO829 truth set

We developed an SV comparison script in R, available at: https://github.com/UMCUGenetics/COLO829_somaticSV/blob/master/SV_benchmarking/SV_benchmarking.R. This script makes use of the StructuralVariantAnnotation package from Bioconductor (v1.2.0) to overlap breakpoints from one or multiple VCFs with a defined truth set. We rerun GRIDSS (v. 2.10) on the ILL data, PBSV (v. 2.4.0) on the PB data and Sniffles (v. 1.0.12) on the ONT data. We compared the original version and the updated version of the calls with the truth set using the script aforementioned. Users can modify overlap parameters like the margin about the breakpoint.

Figure panels 2-A, 3-C and 3D and the graphical abstract were created using Biorender.com.

### QUANTIFICATION AND STATISTICAL ANALYSIS

This study does not involve any statistical analysis or quantification.

