## [Document S2. Transparent peer review records for Valle-Inclan et al · Cell Genomics]

A multi-platform reference for somatic structural variation detection

Jose Espejo Valle-Inclan¹, Nicolle J.M. Besselink¹, Ewart de Bruijn², Daniel L. Cameron^{2,3}, Jana Ebler⁴, Joachim Kutzera¹, Stef van Lieshout², Tobias Marschall⁴, Marcel Nelen⁵, Peter Priestley², Ivo Renkens¹, Margaretha G.M. Roemer⁶, Markus J. van Roosmalen¹, Aaron M. Wenger⁷, Bauke Ylstra⁶, Remond J.A. Fijneman⁸, Wigard P. Kloosterman^{1*}, Edwin Cuppen^{1,2*}

Summary

Initial submission:	Received : September 10, 2020
	Scientific editor: Orli Bahcall and Rita Gemayel
First round of review:	Number of reviewers: 3 Revision invited : January 4, 2021 Revision received : May 6, 2021
Second round of review:	Number of reviewers: 3 Revision invited : August 23, 2021 Revision received : January 24, 2022
Third round of review:	Number of reviewers: 1 Accepted : May 6, 2022
Data freely available:	Yes
Code freely available:	Yes

This transparent peer review record is not systematically proofread, type-set, or edited. Special characters, formatting, and equations may fail to render properly. Standard procedural text within the editor's letters has been deleted for the sake of brevity, but all official correspondence specific to the manuscript has been preserved.

Referees' reports, first round of review

Reviewer #1 (Comments to authors)

In the manuscript, Besselink et al present a comprehensive analysis of somatic structural variations (SVs) found in a cancerous and matching normal cell line of the COLO829 genome. For the analysis, the samples were sequenced with regular and barcoded/linked short reads with Illumina and 10xGenomics platforms, as well as with state of the art long-read sequencing platforms of ONT and PacBio. The datasets have a robust average read-depth coverage of ~40x for every technology, which makes these datasets of great interest and importance to the cancer genomics community for developing and benchmarking SV/SNP/indel somatic inference algorithms. Furthermore, the authors report a "gold truth set" of 68 inferred and validated somatic SVs.

Overall the manuscript is well written and easy to follow with language suitable for experts in the field as well as the broader computational biology and medical communities. The sequencing dataset also represents a great resource for the scientific community and would benefit subsequent development of methods and benchmarking for somatic SV detection. Yet there are several questions about the robustness and quality of the reported findings that must be first addressed:

- * The choice long-read aligners for the ONT and PacBio datasets and the rest of the SV inference pipeline components should be discussed and compared
- * The terminology of what is an SV is not defined, yet is of great importance to understand where the thresholds are set for characteristics like "minimum size" of SVs, how complex SVs are counted/reported (single complex SV vs multiple simpler ones), etc.
- * The distinctions between joint SV calling in normal and cancerous genomes (on a per seq tech basis) vs independent SV inference in normal and cancer genomes can have a serious effect on how the "somatic" SV are reported versus germline variants. This needs to be discussed with results demonstrating the differences/similarities in the obtained designations of somatic/distinct SVs
- * The use of PBSV-only approach for SV inference in PacBio datasets is in contrast with the method-ensemble (NanoSV + Sniffles) used for ONT reads. Furthermore, the reported PBSV version 2.0.1 is VERY outdated (released Sept 2018, with 8 releases since), and is incapable of capturing duplications, and overall brings the validity of PacBio-based SV calls into question
- * The choice of default parameters for SV inference methods may substantially influence the reported results of the analysis, with the example being the minimum number of supporting reads required for an SV to be reported in Sniffles method. Not only does this have an impact on the overall truth set evaluation, but also, and more potently so, in the coverage/purity analysis. Moreover, authors mention that the sensitivity was of great importance for SV inference, and thus default parameters may not be the optimized for this analysis.
- * The authors mention that during manual SV curation they have noticed that some SVs are reported with different terms/nomenclature across different methods, which brings into question the quality of all of the automated comparison results for SV callsets comparisons
- * SURVIVOR is used for SV merging and comparison with a very tight (200bp) threshold, while the standard/recommended, especially for long read SVs, is the 1000. The authors should report on how the results change if the threshold is altered.
- * In the study, GRCh37 is used for the reference despite being several years out of date. Some discussion

on how older reference influence the alignments/SV inference process, especially when then comparing to the SV callset obtained on the GRCh38 reference, is needed

* The authors must include a more detailed report on the numbers of SVs inferred by different methods for different samples at different stages (initial inference, somatic filtration, comparisons, etc). This can provide a reader with a more robust understanding about how the methods and technologies compare on the overall SV inference task as well as in detection of somatic events.

* The reported selection of 88 "high-confidence" SVs for validation, as well as subsequent report of some SVs "having evidence in germline control", and some SVs being considered false positives because of the "noise" in the respective genomic contexts, are ad hoc and not well-defined. This brings into question the underlying choices and methodology utilized at every step of validation process. Also, additional discussion is needed w.r.t. long-read vs short-read alignments in reported "noisy" genomic contexts with low complexity sequence and repeat content. This should not be as problematic for long reads as it does for short reads

* Which parameters were used for BIC-SEQ2 CNV inference, and what is the resolution of obtained CNVs? While the subsequent CNV correlation computation mentions the 1MB fragment size, the usability of such low-resolution CNV calls for SV validation (as reported "characterization of COLO829 somatic SV truth set") is not really suitable, especially since manuscript describes validated somatic SVs of < 100bp. Same is applicable for CNVs inferred with Ginkgo.

* Why was only NanoSV used in SV inference in ONT PCR validation step? Why was a different SV inference method Manta (instead of GRIDDS) used in MiSeq PCR SV inference step? And why was a much tighter threshold of 50bp used for SURVIVOR-based merging of SVs in the validation step?

* More discussion/analysis is needed for the comparison with the SV set from the Arora et al study. Specifically, how do the different comparison approaches affect the results and furthermore, how insertion SVs, that are reported as a 1bp entry w.r.t. to the reference, are overlapped with bedtools in a robust manner?

* How is the SV content for microsatellite filtration determined?

* Were any coverage artifacts observed in the 10x Genomics data, similar to what was reported in the recent report by (Aganezov et al, 2020, Genome Research)

* In the discussion, the authors assert that germline SVs are less randomly distributed than the somatic SVs, which can make it harder for long reads-based pipelines to identify them. However, no data are presented to justify this point, and thus it either requires further analysis or removal from the text.

The issues outlined above have an individual and cumulative effect on the quality of the reported gold truth set of somatic SVs. Given the goal of a carefully curated and validated somatic SV gold truth set, all methodological details need to be explicit and justified.

Reviewer #2 (Comments to authors)

The authors of the manuscript " A multi-platform reference for somatic structural variation detection" describe a new SV call set that was highly curated and validated and is exceptional as it targets somatic mutations. This is very exciting since other truth set SV papers focus on germline only mutations. The authors have used multiple technologies over a cancer cell line to identify the SV and filter candidates down to a reasonable list of diverse types of SV. Furthermore, they describe the vetting process very well in the main text and it is thus easy to follow.

In the following I list my questions and concerns:

1. As you state COLO829 is still evolving. So why was that chosen as a truth set?
2. You need to further give people advice about comparing their calls to yours. This is hugely important as otherwise it is not clear how to use your calls.
3. Thus, are there older Illumina data available that had been sequenced for this cell line? How does their result compare to yours? I think you need to give indications about if these SVs should be there for longer time or relapse again or change in size.
4. I would encourage you to state if the cell line was derived from a female or male. I think it's a male sample.

5. I would recommend to give some stratification about SV type earlier in the manuscript. Especially since the first section is just talking about coverage alterations. Thus I was under the impression you are just talking about CNV events...
6. Have you annotated the somatic SV if they also occur in other data sets (e.g. gnomadSV which also includes cancer samples)
7. Lift over SV to 38..
8. I would have liked to learn a bit more abbot the somatic SV that were identified. You mentioned a few hitting known genes, but are the majority non coding ? Do they follow the expected ratio of events given the chromosome sizes or other interesting patterns?
9. It is interesting to me that PB and ONT agrees less than Illumina and the two technologies (at least from Figure 2B). Given you mentioned centromeric events I would have assumed you get better mapping with ONT + PB.
10. What regions were ignored on the genome? Or did you perform the analysis genome wide ? E.g. GIAB filtered regions that cannot be robustly accessed.

Reviewer #3 (Comments to authors)

The manuscript by Jose Espejo Valle-Inclan et al. "A multi-platform reference for somatic structural variation detection" describes a diligently elaborated resource for benchmarking structural variation detection. The detection of structural variants from NGS data is challenging in germline and even more so in cancer genomes, where rearrangement events are common. Two cell lines (COLO829) that is from a melanoma and corresponding normal have been sequenced deeply using a battery of all possible different next generation sequencing technologies. Multiple bioinformatics tools have been used to establish the somatic structural variants in the pair of genomes. The identified structural variants were then validated using either PCR or capture followed by sequencing. This led to set of 68 somatic bona fide structural variants.

Below a few comments:

- 1) The authors repeatedly mention that a resource that is inexhaustible is of importance. However, once the sequencing of the COLO829 pair has been done it has served its purpose and the sequences from whichever technology are available for people to test their bioinformatics tools. Sequencing this cell line again is only of relevance when a dramatically new sequencing technology is introduced. For the purpose of validating someones ability to carry out, for example, Illumina sequencing it does not bring much. Clearly the ability to detect structural variants is far more dependent on bioinformatics than the sequencing itself.
- 2) Evidently, somatic structural variants are not all that is present in this data. There are SNVs, indels and copy number variants - these have already been shown in these samples in the article of Craig. While Arora et al. showed structural variation in these samples, however with a less comprehensive coverage of technologies. The data from this work adds substantially to the Craig and Arora data. It would be great if all were taken together.
- 3) Craig mentions considerable aneuploidy of this cell cancer cell line. On reading through the Arora article and this work, I am somewhat puzzled by the issue of ploidy. In this work and the Arora article, the cancer genome shows considerable aneuploidy - largely triploid. Going into the Arora article more it turns out that other cell lines used there show a lot of triploid regions. I am not sure how much this reflects a "normal - non-cell line" cancer genome. Even though this cell line is used widely, it might not be the most suitable for the purpose of reflecting a real world situation.
- 4) At least for COLO829BL immortalization needed to be carried out. This is usually done by transformation with EBV. I would have assumed that the incorporation site of the EBV into the genome should be visible and have an impact on the detection of structural variants. Were incorporation sites of the EBV in the COLO829BL observed?
- 5) Here it is reported that there are 20 SVs reported by Arora that are then; after further inspection, partly not detected. It might be a good negative control to try to verify those SVs using PCR or capture followed by sequencing. It might be that the Colo829 is a different batch or passage than the one used by Arora. Whether the SVs were present in the originally taken sample or not, or whether they happen between different passages can not be asserted.
- 6) The authors study the effect of coverage on the ability to call SVs and mention that coverage 30x and 50x showed higher precision than the higher coverage of 98x. I could offer another explanation for this, which is that tools for SV detection were developed using the lower coverages, typically 30x. As a

consequence the tools might function better at that coverage.

7) The authors state that Fujimoto report on widespread indels as a hallmark of microsatellite instable tumors - this was reported earlier by Stobbe et al. in Plos Computational Biology.

8) The study was done using GRCh37, moving forward this might limit the shelf life of this study. It would be good to provide the SV set on GRCh38.

However, what I am mentioning should be considered suggestions rather than major criticism. The authors are making a hugely valuable dataset available to the community to benchmark their bioinformatics tools. This has great merit.

Authors' response to the first round of review

Reviewer #1: In the manuscript, Besselink et al present a comprehensive analysis of somatic structural variations (SVs) found in a cancerous and matching normal cell line of the COLO829 genome. For the analysis, the samples were sequenced with regular and barcoded/linked short reads with Illumina and 10xGenomics platforms, as well as with state of the art long-read sequencing platforms of ONT and PacBio. The datasets have a robust average read-depth coverage of ~40x for every technology, which makes these datasets of great interest and importance to the cancer genomics community for developing and benchmarking SV/SNP/indel somatic inference algorithms. Furthermore, the authors report a "gold truth set" of 68 inferred and validated somatic SVs.

Overall the manuscript is well written and easy to follow with language suitable for experts in the field as well as the broader computational biology and medical communities. The sequencing dataset also represents a great resource for the scientific community and would benefit subsequent development of methods and benchmarking for somatic SV detection. Yet there are several questions about the robustness and quality of the reported findings that must be first addressed:

* The choice long-read aligners for the ONT and PacBio datasets and the rest of the SV inference pipeline components should be discussed and compared:

We added the following text in the results (Page 7):

We used state-of-the-art SV calling tools appropriate for each of the sequencing datasets. Due to the lack of an existing benchmark and best-practices protocols in the somatic SV calling field, this study was oriented to the creation of a gold truth set and not to the benchmarking of somatic SV calling tools and sequencing technologies. Therefore, we chose optimal mapping and SV calling tools to the best of our knowledge and explicitly invite other users to use our data and/or the truth set of SVs for benchmarking of platforms and platform-specific analysis tools that do indeed continuously improve, especially for long read aligners and variant callers.

* The terminology of what is an SV is not defined, yet is of great importance to understand where the thresholds are set for characteristics like "minimum size" of SVs, how complex SVs are counted/reported (single complex SV vs multiple simpler ones), etc.

We added the following text in the results (Page 7):

To avoid inconsistencies derived from nomenclature and classification of SVs in the different datasets, we focused on the detection of individual breakpoints

rather than complex events, with a minimum event size of 30 bp. We used state-of-the-art SV calling tools appropriate for each of the sequencing datasets.

Response to reviewers

* The distinctions between joint SV calling in normal and cancerous genomes (on a per seq tech basis) vs independent SV inference in normal and cancer genomes can have a serious effect on how the "somatic" SV are reported versus germline variants. This needs to be discussed with results demonstrating the differences/similarities in the obtained designations of somatic/distinct SVs. We do fully agree with this point and we will add additional discussion on this topic. It should be noted that for certain data (e.g. ONT and 10x) no somatic callers exist yet (but will likely be developed in the near future as we know from PacBio and Illumina data that such callers are much more powerful than subtraction approaches of two independently called datasets.

We added the following text in the discussion (Pages 15-16):

For example, joint-calling in the tumor and normal sequencing data and cancerspecific somatic filtering is very important to reduce false positive rates in somatic SV calling. However, such an approach was only available for the Illumina dataset, as no somatic-specific callers or protocols exist for the other datasets yet.

* The use of PBSV-only approach for SV inference in PacBio datasets is in contrast with the method-ensemble (NanoSV + Sniffles) used for ONT reads. Furthermore, the reported PBSV version 2.0.1 is VERY outdated (released Sept 2018, with 8 releases since), and is incapable of capturing duplications, and overall brings the validity of PacBio-based SV calls into question.

This is a fair point and reflects the rapid pace by which analysis tools for certain platforms do develop. As indicated above, the primary goal of our study was not to compare software tools or versions thereof, but to generate a truth set that allows for this in a straightforward way. Nevertheless, we took advantage of this suggestion to create a "Benchmarking" result section using an updated version of PBSV, Sniffles and GRIDSS and illustrating the potential use of the SV truth dataset. We also made the updated calls available in a new data bundle.

Benchmarking against the COLO829 truth set:

To aid future benchmarking using the COLO829 truth set described, we developed a script to directly compare SVs to this or other future benchmarks. This script compares SVs at the breakpoint resolution to generate a precision/recall plot. To demonstrate its use, we compared the ILL, ONT and PB calls used for the development of the truth set with the most updated versions of SV variant calling tools available at the time of submission of this work (Supplementary Figure 4). We observed an improvement in recall for the updated version of PBSV. The drop in precision for GRIDSS is likely due to differential manual pre-processing of the original GRIDSS file, which was substituted by automatic filtering in the updated version. Surprisingly, a drop in recall can be observed in the updated version of Sniffles, while maintaining the low precision. We did not further analyse in detail the causes for these changes as this is beyond the scope of the current work, but the framework presented does allow for a versatile approach to bioinformatic tool and parameter optimization. We included the updated VCF files from each technology in the updated data bundle. Any other benchmarking with own VCF files can be performed using the R script with our COLO829 or other future truth sets.

* The choice of default parameters for SV inference methods may substantially influence the reported results of the analysis, with the example being the

minimum number of supporting reads required for an SV to be reported in Sniffles method. Not only does this have an impact on the overall truth set evaluation, but also, and more potently so, in the coverage/purity analysis. Moreover, authors mention that the sensitivity was of great importance for SV inference, and thus default parameters may not be the optimized for this analysis.

This notion is correct and exactly the reason why having truth sets is so important as these can be used to optimize tool settings for specific conditions. We added the following text on the results (page 13):

Another possible explanation is that SV detection tools have been developed and optimized using sequencing depths around 30x and therefore function better at those depths, needing optimized parameters for optimal performance at different sequencing depths.

* The authors mention that during manual SV curation they have noticed that some SVs are reported with different terms/nomenclature across different methods, which brings into question the quality of all of the automated comparison results for SV callsets comparisons

We put this remark in as one should realize that a long-read sequencing technique can report a large insertion as a single event, while short-read technologies do detect this as two break-ends. To circumvent this issue, we have focused on the identification of break-junctions (converted all events into BND events) instead of automated correct event calling and classification. We now tried to explain this in more detail as it should be clear that these differences do not impact on the completeness or quality of the truth set.

We added the following on the results (page 7):

To avoid inconsistencies derived from nomenclature and classification of SVs in the different datasets, we focused on the detection of individual breakpoints rather than complex events, with a minimum size of 30 bp. We used state-of-the-art SV calling tools appropriate for each of the sequencing datasets.

* SURVIVOR is used for SV merging and comparison with a very tight (200bp) threshold, while the standard/recommended, especially for long read SVs, is the 1000. The authors should report on how the results change if the threshold is altered.

This is indeed a rather arbitrary choice and we tested various lengths with limited impact. We now performed the SV merging and filtering analysis with a uniform 1000bp-threshold. This results in 121 additional breakpoints classified as somatic candidates due to support from 2 technologies. Most of these (84) come from the technologies with most original calls and therefore more false positives, ONT and 10X. We evaluated all these calls in the same way as described in the original pipeline and classified 70 as false positives and 51 as germline, with none being somatic when assessing the raw genomic data and the validation data.

We added these analysis results in Supplementary Figure 2 and the following text in the results (page 8):

To corroborate that the breakpoint-merging threshold of 200bp used in our filtering pipeline was not too stringent, we did a re-run the filtering analysis step using 1000bp as a merging threshold, resulting in an extra 121 breakpoints supported by more than 2 technologies. We verified these breakpoints similarly as the original filtering pipeline and classified 70 as false positives and 51 as germline, resulting in no added value for the truth set (Supplementary Figure 2E).

* In the study, GRCh37 is used for the reference despite being several years out of date. Some discussion on how older reference influence the alignments/SV inference process, especially when then comparing to the SV callset obtained on the GRCh38 reference, is needed.

We now provide a GRCh38 somatic SV VCF file in an updated Data bundle, and provide a description of how it was generated in the Methods (Page 23): Liftover to GRCh38

Somatic SV breakpoint positions were lifted over to GRCh38 genomic coordinates using the ENSEMBL Assembly Converter (https://www.ensembl.org/Homo_sapiens/Tools/AssemblyConverter). We used a custom script in order to change the ALT field in the VCF. The lifted-over VCF file is available in the data bundle.

* The authors must include a more detailed report on the numbers of SVs inferred by different methods for different samples at different stages (initial inference, somatic filtration, comparisons, etc). This can provide a reader with a more robust understanding about how the methods and technologies compare on the overall SV inference task as well as in detection of somatic events.

As we discuss in page 15-16: This study was not designed to compare performance of sequencing platforms or data analysis pipelines, since that benchmarking would require the very latest platform, chemistries and pipeline versions to be useful.

Therefore, we argue that this type of comparison would not be relevant within the scope of this manuscript, which is to create a somatic truth set regardless of their methodological origin and to demonstrate its utility. Nevertheless, as all raw and variant call data is publicly available, interested readers can perform such analyses with any tool or setting of choice.

* The reported selection of 68 "high-confidence" SVs for validation, as well as subsequent report of some SVs "having evidence in germline control", and some SVs being considered false positives because of the "noise" in the respective genomic contexts, are ad hoc and not well-defined. This brings into question the underlying choices and methodology utilized at every step of validation process. Also, additional discussion is needed w.r.t. long-read vs short-read alignments in reported "noisy" genomic contexts with low complexity sequence and repeat content. This should not be as problematic for long reads as it does for short reads.

Intuitively, we would agree, but the data tells what the data tells. There are clearly imperfections in both the measurement techniques and the bioinformatics plus effects of sequencing coverage and quality related to primary sequence context. The latter appears, at least in some cases, independent of sequencing methods. We believe that with the multi-platform, independent validation and manual raw data curation approach, we have done the best possible with current technology to come to the most robust SV reference set possible. As discussed, we are still missing a few events that are not detectable with any of the techniques (and existing data analysis tools) but we are sure the 68 SVs reported here are true. To better reflect this, we have added and modified the following discussion regarding low long read contribution:

Likely, this is mostly due to the more advanced somatic SV calling pipelines developed for short-read data than for long-read data, as previously discussed. However, this observation may also be explained by fundamental differences between germline and somatic SVs, such as the overall distribution throughout

the genome, the involvement of repetitive regions and the total amount in such events. With further methodological advances in the somatic and germline SV calling these differences will undoubtedly be further characterized and better understood.

* Which parameters were used for BIC-SEQ2 CNV inference, and what is the resolution of obtained CNVs? While the subsequent CNV correlation computation mentions the 1MB fragment size, the usability of such low-resolution CNV calls for SV validation (as reported "characterization of COLO829 somatic SV truth set") is not really suitable, especially since manuscript describes validated somatic SVs of < 100bp. Same is applicable for CNVs inferred with Ginkgo.

Unfortunately, not a single CNV calling tool could be used for all data types, but we used this data only to identify break-junctions that are apparently missing from our truth set. Furthermore, the CNA comparison at low resolutions was used to demonstrate that all methods yielded highly similar results at least at relatively low resolution. We have not attempted to optimize or benchmark CNV calling tools, although with the truth set generated, this is quite simple to do for any tool. It should be noted that each CNA should be associated with two BND SV events. We will better explain the meaning and relevance of the CNV figure in the manuscript and add the disclaimer that the approach used by us is only suited for detecting and comparing large events.

We modified the following text in the results (page 5):

Unfortunately, no single CNA calling was available to detect CNAs with high resolution for all technologies. Nevertheless, low resolution CNA calling revealed a highly similar copy number profile for each of the technologies.

And in the methods (page 21):

Somatic CNA calling was performed on the ILL dataset with BIC-SEQ2 with default parameters (v0.7.2, (Xi et al. 2016)). For the remaining datasets, since no specific genome-wide CNA calling algorithms were available for each technology, BAM and optical map (xmap) files were converted to BED format using Bedtools (v2.25.0, (Quinlan and Hall 2010)) and CNA calling was performed with Ginkgo (Garvin et al. 2015).

* Why was only NanoSV used in SV inference in ONT PCR validation step?

Why was a different SV inference method Manta (instead of GRIDDS) used in MiSeq PCR SV inference step? And why was a much tighter threshold of 50bp used for SURVIVOR-based merging of SVs in the validation step?

The methodological choices were made for practical reasons, which we describe in more detail in the methods. In all cases, we have used relatively loose criteria to not miss any real event. We have not attempted to optimize any method as our primary focus was to generate a truth set of SVs at nucleotide resolution (hence the tighter threshold for the validation step as this should validate the exact event, while merging events obtained from different technologies or pipelines could be impacted by the pipeline output formats).

We added the following text in the methods (page 22):

The threshold to merge SVs was tighter in the validation dataset than in the raw genomic dataset due to the highly targeted approach and the small size of a few base pairs of the amplicons.

* More discussion/analysis is needed for the comparison with the SV set from the Arora et al study. Specifically, how do the different comparison approaches affect the results and furthermore, how insertion SVs, that are reported as a 1bp entry w.r.t. to the reference, are overlapped with bedtools in a robust manner?

In our analyses, we have converted all events to BND events before comparison to excluded undesired impact of annotation differences such as those mentioned here. So, insertions are also treated as BNDs and can be compared in a robust manner.

* How is the SV content for microsatellite filtration determined?

Microsatellite instability is not classified as an SV but falls into the indel category and is thus not considered.

We added the following text in the methods (page 23):

SVs involving unstable microsatellites were not considered as part of our analyses

* Were any coverage artifacts observed in the 10x Genomics data, similar to what was reported in the recent report by (Aganezov et al, 2020, Genome Research)

We analyzed the coverage artifacts from Aganezov et al. No somatic SV described in our truth set falls onto the 7228 abnormal coverage regions described there. Furthermore, when applying similar filter as in Aganezov et al., only 279 out of those 7228 regions show abnormal coverage in our dataset, with the following distribution:

Nevertheless, we think that this analysis is out of the scope of the manuscript and have not included these results in the manuscript.

* In the discussion, the authors assert that germline SVs are less randomly distributed than the somatic SVs, which can make it harder for long reads-based pipelines to identify them. However, no data are presented to justify this point, and thus it either requires further analysis or removal from the text.

Analyses have shown that repeat involvement is different between germline and somatic SVs, although it remains unclear if differences in mechanism or selection is underlying this.

Nevertheless, we have modified the relevant section in the discussion as we agree most of this speculative:

Likely, this is mostly due to the more advanced somatic SV calling pipelines developed for short-read data than for long-read data. However, this observation may also be explained by fundamental differences between germline and somatic SVs, such as the overall distribution throughout the genome, the involvement of repetitive regions and the total amount of such events. With advances in the somatic and germline SV calling these differences will undoubtedly be further characterized and better understood.

The issues outlined above have an individual and cumulative effect on the quality of the reported gold truth set of somatic SVs. Given the goal of a carefully curated and validated somatic SV gold truth set, all methodological details need to be explicit and justified.

We do fully agree. Each variant classified in the truth set should indeed be true. This does not mean that we want to make any claim about detectability by any given technology. We believe the revisions to the manuscript as a response to the various reviewers' comments do now better reflect this main focus of our work.

Reviewer #2: The authors of the manuscript " A multi-platform reference for somatic structural variation detection" describe a new SV call set that was highly curated and validated and is exceptional as it targets somatic mutations. This is

very exciting since other truth set SV papers focus on germline only mutations. The authors have used multiple technologies over a cancer cell line to identify the SV and filter candidates down to a reasonable list of diverse types of SV. Furthermore, they describe the vetting process very well in the main text and it is thus easy to follow.

In the following I list my questions and concerns:

1. As you state COLO829 is still evolving. So why was that chosen as a truth set?

We will rephrase this. COLO829 is genetically relatively stable, but we and others have shown that any cell line, primary culture and even cell in an organism is subject to mutation accumulation, which is intrinsic to life due to genotoxic exposure and DNA replication errors during cell division. We rephrased this section and added the notion that novel variants not present in current truth set should be validated independently as these could be due to inevitable cell line evolution.

We added the following text in the discussion (page 17):

While the COLO829 cell line is relatively stable genomically, it has a certain level of genetic heterogeneity and is subject to mutation accumulation and evolution throughout culture like any cell line. This variation is dynamic and might differ between cell line isolates as already demonstrated by the various studies on this cell line (Velazquez-Villarreal et al. 2020; Craig et al. 2016) and thus limit the utility of a single defined truth set obtained as presented here. Therefore, novel somatic SVs not present in our current truth set should be validated independently.

2. You need to further give people advice about comparing their calls to yours. This is hugely important as otherwise it is not clear how to use your calls.

This is a good suggestion. We added a paragraph in the results about benchmarking using our truth set (page 14) and updated version of somatic SV tools. We made a script available to compare other somatic SV calls to our truthset, providing recall and precision metrics:

Benchmarking against the COLO829 truth set:

To aid future benchmarking using the COLO829 truth set described, we developed a script to directly compare SVs to this or other future benchmarks. This script compares SVs at the breakpoint resolution to generate a precision/recall plot. To demonstrate its use, we compared the ILL, ONT and PB calls used for the development of the truth set with the most updated versions of SV variant calling tools available at the time of submission of this work (Supplementary Figure 4). We observed an improvement in recall for the updated version of PBSV. The drop in precision for GRIDSS is likely due to differential manual pre-processing of the original GRIDSS file, which was substituted by automatic filtering in the updated version. Surprisingly, a drop in recall can be observed in the updated version of Sniffles, while maintaining the low precision. We did not further analyse in detail the causes for these changes as this is beyond the scope of the current work, but the framework presented does allow for a versatile approach to bioinformatic tool and parameter optimization. We included the updated VCF files from each technology in the updated data bundle. Any other benchmarking with own VCF files can be performed using the R script with our COLO829 or other future truth sets.

3. Thus, are there older Illumina data available that had been sequenced for this cell line? How does their result compare to yours? I think you need to give indications about if these SVs should be there for longer time or relapse again

or change in size.

We do this already with the comparison with the Arora et al dataset. Clearly the cell lines evolved in the time between their use in both labs (our clone was a freshly obtained vial from ATCC) but are at the structural level largely the same with only a few novel events. We have now added a clear disclaimer to the text regarding the use of different batches of cells for experimental benchmarking. This inevitable cell line evolution is of course not relevant when benchmarking SV calling tools but only for benchmarking experimental methods.

We modified and added the following text in the discussion (page 16-17):

Apart from the benchmarking opportunities provided by our truth set, the COLO829 cell line has the advantage that it is, in contrast to real tumor samples, a renewable source. Therefore, it can be used for assessing the impact of future platform developments or the performance of completely new technologies for somatic mutation detection by generating new datasets from the same cell line. [...]

Therefore, novel somatic SVs not present in our current truth set should be validated independently, especially when data was generated using different batches of cells from the COLO829 cell lines.

4. I would encourage you to state if the cell line was derived from a female or male. I think it's a male sample.

We added the following text in the introduction (page 4):

These cell lines were derived from a male individual and have been used before to establish somatic SNV and copy number alteration (CNA) reference sets.

5. I would recommend to give some stratification about SV type earlier in the manuscript. Especially since the first section is just talking about coverage alterations. Thus I was under the impression you are just talking about CNV events...

Thanks for the suggestion. We added the following sentence to start the results section (page 5):

In this study, we aimed to obtain a high-quality validated set of somatic structural variations.

6. Have you annotated the somatic SV if they also occur in other data sets (e.g. gnomadSV which also includes cancer samples)

This is an interesting suggestion. We overlapped the somatic SV truth set with gnomad-SV and with the segmental duplications, simple repeats and microsatellite annotations, but this did not reveal a relevant overlap. We included this information in the results (page 10):

Annotation of the somatic SV breakpoints with gnomAD-SV (Collins et al. 2020), segmental duplications, simple repeats or microsatellites from the UCSC genome browser did not reveal any overlap.

7. Lift over SV to 38..

This information is now added. See reviewer #1.

8. I would have liked to learn a bit more abbot the somatic SV that were identified. You mentioned a few hitting known genes, but are the majority non coding ? Do they follow the expected ratio of events given the chromosome sizes or other interesting patterns?

Apart from the annotations described in point 7, we added the following in the results (page 10):

There are breakpoints in all chromosomes except 2, 13, 17 and 21 (Figure 3B). These chromosomes also do not show any CNA event. Annotation of the somatic SV breakpoints with gnomAD-SV (Collins et al. 2020), segmental

duplications, simple repeats or microsatellites from the UCSC genome browser did not reveal any overlap.

9. It is interesting to me that PB and ONT agrees less than Illumina and the two technologies (at least from Figure 2B). Given you mentioned centromeric events I would have assumed you get better mapping with ONT + PB.

We agree that the data from the different platforms reveals interesting and unexpected patterns. However, it is not our aim to compare performance of different methods, also because this would require the very latest platforms and chemistries to be useful. As indicated above, we have focused on not missing any real event to come to a truth set. To make this even more clear in the revised manuscript we added the following sentence in the discussion (page 15):

This study was not designed to compare performance of sequencing platforms or data analysis pipelines, since that benchmarking would require the very latest platform, chemistries and pipeline versions to be useful.

10. What regions were ignored on the genome? Or did you perform the analysis genome wide ? E.g. GIAB filtered regions that cannot be robustly accessed.

The analysis was performed genome wide as accessibility is technology dependent and we did not want to miss any real event. We added the following text in the Methods (Page 21):

All analyses aforementioned were performed genome-wide and no genomic regions from GRCh37 were filtered or ignored.

Reviewer #3: The manuscript by Jose Espejo Valle-Inclan et al. "A multiplatform reference for somatic structural variation detection" describes a diligently elaborated resource for benchmarking structural variation detection. The detection of structural variants from NGS data is challenging in germline and even more so in cancer genomes, where rearrangement events are common. Two cell lines (COLO829) that is from a melanoma and corresponding normal have been sequenced deeply using a battery of all possible different next generation sequencing technologies. Multiple bioinformatics tools have been used to establish the somatic structural variants in the pair of genomes. The identified structural variants were then validated using either PCR or capture followed by sequencing. This led to set of 68 somatic bona fide structural variants.

Below a few comments:

1) The authors repeatedly mention that a resource that is inexhaustible is of importance. However, once the sequencing of the COLO829 pair has been done it has served its purpose and the sequences from whichever technology are available for people to test their bioinformatics tools. Sequencing this cell line again is only of relevance when a dramatically new sequencing technology is introduced. For the purpose of validating someones ability to carry out, for example, Illumina sequencing it does not bring much. Clearly the ability to detect structural variants is far more dependent on bioinformatics than the sequencing itself.

We agree that the largest value of a truth set is in bioinformatic pipeline development and parameter optimization (e.g. at different sequencing depths). However, we believe both the cell line and the SV truth set remain highly valuable to benchmark new sequencing technologies as well, as one expects the vast majority of SVs described in the truth set to be detectable, unless a new event is observed that can explain disappearance of the anticipated events.

Newly observed events could be due to cell line batch effects or evolution, and should be independently validated. From comparison of the Arora data and our own data, it is not expected that there are typically more than 10% more novel events, though.

We modified and added the following sentences to the discussion to better reflect the utility of the resource (page 16):

Apart from the benchmarking opportunities provided by our truth set, the COLO829 cell line has the advantage that it is, in contrast to real tumor samples, a renewable source. Therefore, it can be used for assessing the impact of future platform developments or the performance of completely new technologies for somatic mutation detection by generating new datasets from the same cell line.

2) Evidently, somatic structural variants are not all that is present in this data. There are SNVs, indels and copy number variants - these have already been shown in these samples in the article of Craig. While Arora et al. showed structural variation in these samples, however with a less comprehensive coverage of technologies. The data from this work adds substantially to the Craig and Arora data. It would be great if all were taken together.

It is correct that SNVs and indels have been described in detail previously. We have focused specifically on SVs as these were still the types of somatic variants for which no good truth set exists and which are also most challenging to detect. Furthermore, we do not believe that long-read technologies do add to SNV and indel detection in a study that aims for generating truth sets. We believe including SNV and indel benchmarks to this manuscript would deviate from the main scope and attract even more attention to platform comparison, which is not our intention.

3) Craig mentions considerable aneuploidy of this cell cancer cell line. On reading through the Arora article and this work, I am somewhat puzzled by the issue of ploidy. In this work and the Arora article, the cancer genome shows considerable aneuploidy - largely triploid. Going into the Arora article more it turns out that other cell lines used there show a lot of triploid regions. I am not sure how much this reflects a "normal - non-cell line" cancer genome. Even though this cell line is used widely, it might not be the most suitable for the purpose of reflecting a real world situation.

We agree that a cell line is not the same as a tumor sample and this statement is also clearly present in the manuscript (including the statement that truth sets should in the future be expanded to more cell lines and larger SV landscape diversity). However, for SV benchmarking this cell line is still extremely valuable as it assesses all types of SV events including the presence of aneuploidies (which is a very common tumor characteristic and clearly detected in COLO829 in our dataset as well).

4) At least for COLO829BL immortalization needed to be carried out. This is usually done by transformation with EBV. I would have assumed that the incorporation site of the EBV into the genome should be visible and have an impact on the detection of structural variants. Were incorporation sites of the EBV in the COLO829BL observed?

Detection of SV events specific to the control samples requires a different analysis than done here as they can not be detected with standard somatic variant callers using tumor vs normal as comparison. This would require 'standard' germline SV calling on the BL sample. Although we could simply do this, we believe this is out of the scope of the manuscript since we focused on

somatic variant calling.

5) Here it is reported that there are 20 SVs reported by Arora that are then; after further inspection, partly not detected. It might be a good negative control to try to verify those SVs using PCR or capture followed by sequencing. It might be that the Colo829 is a different batch or passage than the one used by Arora. Whether the SVs were present in the originally taken sample or not, or whether they happen between different passages can not be asserted.

This is an interesting suggestion, but without access to material from both batches, this experiment cannot be executed properly. Furthermore, we believe this would be a lot of effort for demonstrating something we already know (and discuss in the paper); cell line evolution. In addition, addressing this issue for these two studies does not have any added value for any future experiment or study or the truth set described here. We are therefore not very eager pursuing this route experimentally.

6) The authors study the effect of coverage on the ability to call SVs and mention that coverage 30x and 50x showed higher precision than the higher coverage of 98x. I could offer another explanation for this, which is that tools for SV detection were developed using the lower coverages, typically 30x. As a consequence the tools might function better at that coverage.

Good point. We added the following sentence to the results (page 13):

Another possible explanation is that SV detection tools have been developed and optimized using sequencing depths around 30x, and therefore function better at those parameters.

7) The authors state that Fujimoto report on widespread indels as a hallmark of microsatellite instable tumors - this was reported earlier by Stobbe et al. in Plos Computational Biology.

Thanks for noting. We replaced this reference.

8) The study was done using GRCh37, moving forward this might limit the shelf life of this study. It would be good to provide the SV set on GRCh38.

GRCh38 coordinates have now been included (see reviewer #1).

However, what I am mentioning should be considered suggestions rather than major criticism. The authors are making a hugely valuable dataset available to the community to benchmark their bioinformatics tools. This has great merit.

Referees' report, second round of review

Reviewer #1 (Comments to authors)

The authors have addressed all of my concerns, and the manuscript is ready for publication.

Reviewer #2 (Comments to authors)

I would like to thank the authors to comment on all my questions from the previous review round. However, I still have several concerns about this work .

1. You need to quantify what "but are at the structural level largely the same with only a few novel events" means. Since this is a gold standard truth set that you put out there... Your disclaimer just highlights the non-usable for this data set and work as a benchmark set. Thus, how is this truthset then different from e.g. downloading TCGA data and running it?

2. The answer of yours to my previous 9th question is very unsatisfactory. You provide here a gold standard benchmark set and make comparisons across callers (implicit) , but state that you don't want to dive in, because the technologies don't agree very well. In the same moment, you require us to believe in your call

set. I agree that you performed several validations, but the question about how valid , comprehensive this benchmark set of yours is if fully unanswered. "This study was not designed to compare performance of sequencing platforms or data analysis pipelines, since that benchmarking would require the very latest platform, chemistries and pipeline versions to be useful." Thus, what do you think this study or your benchmark set should be used for? As you state not for sequencing tech comparison, nor for software methods..

3. What is the value of a benchmark that continue to evolve and differentiate? Is there data showing that these somatic SV that you propose as benchmark are stable?

4. You cannot guarantee you missed somatic SV , you did however do a great job in validating the few that you report. So in best one can assess recall, but not precision right ?

5. In the end you refer to the SV callers that have been run with low precision, but in the beginning you say you ran then in high sensitivity settings? Isn't that then expected?

6. You explicitly mention that this study is providing a gold truth set, but how can you do that if you include the entire genome? Have the centromeres , paracentrones, telomers been accessed? Have you built a truth set across HLA region ? How does this compare to the Chm13 genome ?

7. Did you compare your approach to assembly approaches ?

8. What is your estimate on accuracy on the breakpoints ? Do these different methods agree or not ?

9. What are the minimum length of SV you target here ?

10. You used copy number calling to establish the completeness of your somatic SV calls ? How can/should copy number calling improve the confidence on insertions, or rearrangements? Especially since CNA calling algorithms are know to be imprecise and noisy (e.g. Illumina) . And there are non-existing or benchmarked on long reads.

11. You need to explain how you compare your SV. Just checking breakpoints is not good enough. Is that a bcftools merge ? There are now multiple methods to compare SV that are much more sophisticated and its shown that this is needed.

Reviewer #3 (Comments to authors)

no additional comments

Authors' response to the second round of review

Reviewer #1: The authors have addressed all of my concerns, and the manuscript is ready for publication.

Reviewer #2: I would like to thank the authors to comment on all my questions from the previous review round. However, I still have several concerns about this work .

1. You need to quantify what "but are at the structural level largely the same with only a few novel events" means. Since this is a gold standard truth set that you put out there... Your disclaimer just highlights the non-usable for this data set and work as a benchmark set.

Thus, how is this truthset then different from e.g. downloading TCGA data and running it?

The citation of the reviewer is from our rebuttal letter and is not as such in the manuscript.

A detailed list of unique events from the Aurora manuscript and annotation is present in Supplementary table 5 and absolute numbers are discussed in the main text: *'We found that 58 (75.34%) and 59 (78.6%) of the somatic SV calls for the HiSeq and the NextSeq callsets, respectively, overlapped with our somatic SV truth set on both sides of the SV*

(Supplementary Figure 3). We manually inspected the 20 non-overlapping somatic SV calls from the Arora et al dataset in our raw ILL, ONT and PB data (Supplementary Table 5). In the long-read raw data (ONT and PB) only 3 out of the 20 have some support (maximum 3

reads). In the ILL raw data, 9 out of the 20 have limited evidence, with only one or a few supporting reads. Only 4 of these 9 SV calls passed bioinformatic calling criteria in our original ILL somatic SV calls, but none of these were called by any other technology or independently validated by more sensitive PCR or targeted capture and deep-sequencing. Therefore, we consider these candidates as technology-specific noise and were discarded from our truth set, although we can formally not exclude that these are real variants that are present at very low frequency (<1% in the sample). Finally, 13 SVs are present in our truth set and not in the Arora et al. data set. All were detected by at least two different sequencing techniques and independently validated.'

Our truth set is different from TCGA data as events in the TCGA dataset are not based on multiple technologies and are not independently validated and thus likely include more false negatives and positives, respectively.

2. The answer of yours to my previous 9th question is very unsatisfactory. You provide here a gold standard benchmark set and make comparisons across callers (implicit), but state that you don't want to dive in, because the technologies don't agree very well. In the same moment, you require us to believe in your call set. I agree that you performed several validations, but the question about how valid, comprehensive this benchmark set of yours is if fully unanswered. "This study was not designed to compare performance of sequencing platforms or data analysis pipelines, since that benchmarking would require the very latest platform, chemistries and pipeline versions to be useful." Thus, what do you think this study or your benchmark set should be used for? As you state not for sequencing tech comparison, nor for software methods..

The statement of the reviewer that 'the technologies don't agree very well' contrasts with what we write in the discussion: 'there is clear complementarity between the various platforms for the comprehensive identification of all real events.'

Regarding the question 'what do you think this study or your benchmark set should be used for', we have included the 'Benchmarking against the COLO829 truth set' section at the end of the results section. This includes an illustration of the utility of the truth set and includes the required software tools to repeat such analysis for own datasets.

3. What is the value of a benchmark that continue to evolve and differentiate? Is there data showing that these somatic SV that you propose as benchmark are stable?

The comparison with the Arora et al data shows that this cell line is very stable. Of course, a cell line will always evolve as any piece of living material does (like commonly used biomedical models, e.g. strains of mice or HeLa cells or an organoid), so a perfect alternative for a living renewable system does not exist as far as I am aware.

However, to better highlight these limitations, we have added/adapted the following section in the discussion: 'The COLO829 truth set should therefore be used with caution and analyzing additional cancer cell lines with matching normal cell lines may provide an attractive route for future improvements as these represent in principle an endless source of genomic material for benchmarking of future DNA analysis technologies, but also for quality monitoring in routine production labs under accreditation. However, availability of suited cell lines that represent the full genetic diversity of cancer is a clear limitation. Ideally, one would thus resort to thoroughly analysed real tumor samples, even though in practice availability of sufficient material for multi-lab and multi-technology analyses can be problematic and

sharing and reusing of patient material and data may require complex consenting and legal procedures. The use of synthetic samples could also be a complementary approach (Ewing et al. 2015), although it's utility for mimicking complex structural variation remains to be demonstrated and technical challenges may arise when the sequencing technology that one wants to benchmark requires input of high molecular weight molecules.'

4. You cannot guarantee you missed somatic SV , you did however do a great job in validating the few that you report. So in best one can assess recall, but not precision right ? This is correct. We have added the following text to the discussion to make this clear to the readers: *'Of course, we can formally not exclude that there are more events missing from our truth set due to limitations in current sequencing and data analysis approaches for example due to inaccessibility of centromeres, telomeres or other repetitive elements. Therefore, we do recommend following up promising novel candidates that emerge in future benchmarking studies, with orthogonal validations to further improve the current truth set.'*

5. In the end you refer to the SV callers that have been run with low precision, but in the beginning you say you ran then in high sensitivity settings? Isn't that then expected? That is indeed correct. We did run the callers in such a way that we would not miss out on any potential real candidate. Only the Illumina pipelines have been strongly optimised over the past years and that is why that platform has much higher precision in figure 4A than the other technologies. This is not a reflection of the capabilities of the platforms themselves, but illustrates the degree of maturity of the somatic data analysis platforms (which can be further optimised using gold reference sets like the one presented here). This argumentation is already in the manuscript.

6. You explicitly mention that this study is providing a gold truth set, but how can you do that if you include the entire genome? Have the centromeres , paracentrones, telomers been accessed? Have you built a truth set across HLA region ? How does this compare to the Chm13 genome ?

We are indeed blind to events that are missed by current technologies. This limitation is discussed. See also comment #4. In the conclusion, we also discuss missing events that can be inferred from the data but were not detected (breakjunctions that must reside in centromeric regions).

7. Did you compare your approach to assembly approaches ?

This is an interesting suggestion but beyond the scope of the current work. All data is publicly available, so others (who likely have much more experience in this relatively young and specialistic area) can embark on this challenge (which as far as I know has never been done for a cancer genome using multi-platform sequencing data as input).

8. What is your estimate on accuracy on the breakpoints ? Do these different methods agree or not ?

All break-junctions are called and independently verified and curated with nucleotide level accuracy (Supplementary Table 3). Different technologies and platforms do unfortunately output 'raw' calls in diverse formats, which makes it hard to directly compare methods. For example, a short read platform can call a 1 kb insertion as two independent break junctions, while a long-read platform can report such an event as an insertion. For the truth set, input from various methods were manually curated to end up with the nucleotide resolution reference set.

9. What are the minimum length of SV you target here ?

There is not a clear definition on what is a structural or small variant (like an INDEL). However, we used an arbitrary cut-off of 25 bp (for events that are not copy number neutral). We have added this detail to the Methods section.

10. You used copy number calling to establish the completeness of your somatic SV calls ?

How can/should copy number calling improve the confidence on insertions, or rearrangements? Especially since CNA calling algorithms are known to be imprecise and noisy (e.g. Illumina) . And there are non-existing or benchmarked on long reads.

The principle used here is that by definition each copy number alteration should have a SV break-junction at either end. This means that noisy false positive CNA calls do not have associated SV calls and that real CNA's should have 2 SVs associated unless the CNA extends towards a telomere. Indeed, we report and discuss CNAs with a missing SV event that likely resides in a centromere based on copy number differences of the chromosome arms. Long reads also failed to capture break-junction evidence for these events (at least with the currently used bioinformatic data analysis tools).

11. You need to explain how you compare your SV. Just checking breakpoints is not good enough. Is that a bcftools merge ? There are now multiple methods to compare SV that are much more sophisticated and its shown that this is needed.

We used the published tool SURVIVOR for this purpose. This is detailed in the methods section.

Reviewer #3: no additional comments

Referees' report, third round of review

Reviewer #1 (Comments to authors)

I have no further concerns. In support of the authors, the Genome-In-A-Bottle benchmarks have demonstrated enormous value over the year even though they exclude certain types of complex events and exclude certain complicated regions of the genome. I expect the current work will follow a similar highly valuable path